# Non-loss engraved circuit patterning method of semi-liquid metal for precision recyclable multi-substrate circuits

Xiaoqing Li[1,2], Tianyu Li[2], Yubing Liu[1], Chengjie Jiang[2], Yiyi Chen[2], Hui Zong[2], Zihang Zhang[2], Jianye Gao ®[3], Jing Liu ®[1] ✉ & Rui Guo ®[1,2] ✉

Room-temperature liquid metal alloys have emerged as promising materials for flexible electronics due to their unique fluidity, conductivity, and biocompatibility. However, traditional patterning techniques for liquid metal circuits, including additive and subtractive manufacturing, face challenges such as high costs, complex processes, and environmental issues, limiting their large-scale application. This study presents a non-loss method for fabricating high-precision semi-liquid metal circuits by leveraging ethanol to modulate interfacial adhesion between liquid metal and substrates. By precisely controlling adhesion through a custom-designed displacement apparatus, the approach enables seamless patterning from 5 μm to centimeter scales across diverse substrates with features like stretchability (1000% strain), reusability, and recyclability. The technique overcomes limitations of conventional methods, offering advantages in cost-effectiveness, operational simplicity, and substrate compatibility. Demonstrations include multifunctional flexible circuits for wearable electronics, aerospace, and smart home applications, highlighting its potential to advance sustainable, scalable liquid metal electronics manufacturing.

In recent years, room-temperature liquid metal alloys based on low-melting-point gallium have exhibited unique advantages in flexible electronics due to their excellent room-temperature fluidity, high electrical conductivity, solid-liquid phase transition properties, and biocompatibility[1–3]. These gallium-based alloys, integrating metallic and liquid properties, are widely employed in multifunctional flexible electronics and have garnered significant attention across disciplines including electronic skin[4], soft robotics[5], health monitoring[6], information storage[7,8], and energy systems[9].

As a liquid, liquid metal exhibits infinite deformability[10] and self-healing capacity[11]. The infinite deformability of liquid metal enables it to adapt to the mechanical properties of various flexible substrates, including high-molecular polymers, paper, fabrics and skin, and makes it applicable for the fabrication of wearable electronic devices on multiple substrates[12,13]. Stretchable interconnection wires made of liquid metal can be used to connect rigid devices attached to flexible substrates, forming an island-bridge structure, which can be applied to fabricate conformal circuits on complex curved surfaces, such as electrochemical sensors attached to the skin surface and conformal electrode arrays on the surface of internal organs[14–16]. However, the high fluidity and surface tension of liquid metal pose challenges for patterning via traditional circuit printing techniques[17,18]. To fabricate high-precision liquid metal circuits, processing techniques such as photolithography and ion-beam etching[19–21], which are costly and time-consuming, are often required. This severely restricts the widespread use of liquid metal flexible circuits. In addition, the substrate materials used in various flexible electronic devices vary significantly in performance and surface morphology. Therefore, it is necessary to develop

[1]State Key Laboratory of Cryogenic Science and Technology, Technical Institute of Physics and Chemistry, Chinese Academy of Sciences, Beijing, China. [2]School of Precision Instrument and Opto-Electronics Engineering, Tianjin University, Tianjin, China. [3]Department of Biomedical Engineering, School of Medicine, Tsinghua University, Beijing, China. ✉e-mail: jliu@mail.ipc.ac.cn; guorui@mail.ipc.ac.cn

liquid metal circuit manufacturing methods that match the interfacial characteristics of the substrates.

To address the above issues, various liquid metal patterning techniques based on different principles have been developed in recent years, which mainly include two forms: additive manufacturing and subtractive manufacturing. Additive manufacturing refers to techniques based on material accumulation principles. These include: direct writing or 3D printing using surface-modified liquid metal[22–24]; screen printing or transfer printing utilizing customized templates[25–30]; direct fabricating the target circuit patterns leveraging laser engraving technology on specific substrates that need to be laser-activated[31–33] and microchannel perfusion[34–37], among others. Nevertheless, the application of additive manufacturing for fabricating liquid metal circuits typically incurs substantial costs and stringent processing demands. This is attributable to the requirement for customized nozzles, precision templates, specialized printing apparatus adept at fabricating intricate microchannels, and, in some cases, laser systems. Subtractive manufacturing represents a material-removal-based approach for circuit fabrication. For instance, it involves initially creating metal circuit patterns via laser sintering or chemical selective deposition, followed by methods such as contact wetting with liquid metal[38,39]. Despite its utility, subtractive manufacturing confronts notable challenges. This technique predominantly depends on costly photolithography apparatus. During the circuit fabrication process, it entails operations like pasting and peeling. These not only substantially elevate equipment expenses but also generate waste, giving rise to environmental contamination and further augmenting circuit manufacturing costs. Conventional liquid metal patterning techniques, encompassing both additive and subtractive manufacturing approaches, are encumbered by significant drawbacks, including elevated costs, intricate procedures, and environmental pollution. These limitations pose substantial barriers to the advancement and large-scale implementation of liquid metal-based flexible electronics technology. Consequently, the development of a liquid metal patterning methodology characterized by low cost, streamlined processing, high precision, compatibility with diverse substrates, zero waste production, and recyclability of raw materials has become an imperative research priority.

Building on our previous research on the selective adhesion mechanism of liquid metal[40,41], we discovered that ethanol can precisely regulate the interfacial adhesion between semi-liquid metal and substrates. Leveraging this discovery, we demonstrate that by precisely applying ethanol to control the semi-liquid metal's adhesion at designated locations on the substrate, it becomes feasible to achieve circuit patterning with high precision. Compared with previous studies where hydrogen bonds serve to enhance the adhesion between liquid metals and substrates, this study provides an approach that utilizes hydrogen bonds in ethanol to hinder the re-adhesion of semi-liquid metals to substrates. In response to the challenges of intricate operation, elevated costs, and resource inefficiency inherent in conventional high-precision flexible electronics manufacturing techniques, our study presents an innovative non-loss engraved circuit patterning (NECP) method that diverges significantly from traditional subtractive and additive methodologies. This process is characterized by zero material wastage, and it proficiently satisfies the stringent technical demands of high precision, compatibility with diverse substrates, stretchability, reusability, and recyclability. Notably, it offers a distinct advantage in enabling the cost-effective, large-scale production of flexible circuits. This study elucidates the mechanism through which ethanol modulates the adhesion of semi-liquid metal to substrates. We designed and fabricated a high-precision displacement apparatus, enabling precise manipulation of the interfacial adhesion between liquid metal and substrates at targeted locations. Leveraging this method, we successfully achieved patterning of semi-liquid metal across diverse substrates and demonstrated the fabrication of flexible circuits spanning multiple scales, from micrometers to centimeters.

This non-destructive patterning technology is characterized by high efficiency, operational simplicity, and customizable precision, offering broad application prospects in wearable electronics, aerospace, and smart home systems.

## Results

### Principles and advantages of the NECP method

In this study, we obtained semi-liquid metal by doping solid metal particles into liquid metal (Supplementary Fig. 1), which exhibits lower fluidity compared to liquid metal. Numerous previous studies[42,43] have confirmed that metallic silver exhibits better wettability with liquid metals than metallic copper, and relevant characterizations have revealed the presence of intermetallic compounds. Therefore, to facilitate the fabrication of semi-liquid metals and reduce material costs, we have selected metal particles with a silver coating on the surface of copper particles. Such metal particles are significantly cheaper than silver particles and still possess wettability similar to that of silver particles. This reduced fluidity overcomes the poor printability of liquid metal with various substrates due to its high surface tension, enabling the semi-liquid metal to be uniformly coated on the surfaces of multiple substrates. By preparing a series of samples with doping ratios of 0 wt%, 5 wt%, 10 wt%, 15 wt%, 20 wt%, and 25 wt%, we found that when the doping ratio of silver coated copper particles reaches 25 wt%, the physical form of semi-liquid metal becomes powdery, which cannot be used in printed circuits, and when the doping ratio above 25 wt%, the particles are difficult to enter the liquid metal, and the doping amount of silver-plated copper particles is negatively correlated with the fluidity of the semi-liquid metal (Supplementary Fig. 2). The results in Supplementary Fig. 3 demonstrate an increase in electrical conductivity as the doping ratio escalates from 0 wt% to 20 wt%. When the doping ratio was 15 wt%, the semi-liquid metal presented a slurry suitable for brushing and a higher electrical conductivity of $9 \times 10^6$ S/m, therefore subsequent studies were all based on this ratio. Figure 1a shows a functional schematic of fabricating semi-liquid metal circuits using the NECP method. This method achieves patterning through the synergistic regulation of the interfacial adhesion between semi-liquid metal and flexible substrates by a needle and ethanol. Silver coated copper particles exhibits excellent corrosion resistance, which ensures the long-term stability of semi-liquid metal. The Scanning Electron Microscope (SEM) photos in Supplementary Fig. 4 show the micro-morphology of the semi-liquid metal film prepared two months ago. This figure clearly indicates that only a slight change has occurred in the diameter of the solid particles. The X-ray Photoelectron Spectroscopy (XPS) curves in Supplementary Fig. 5 indicate that the surface of the semi-liquid metal is encapsulated by a metal oxide film predominantly composed of gallium oxide (Gallium oxide, $Ga_2O_3$). As a result, its physical state can remain relatively stable, and it exhibits long-term mechanical stability, maintaining good stretchability even after long-term storage. Prior investigations have established that the hydrogen bond interaction between this oxide film and the flexible substrate constitutes the primary mechanism driving adhesion. In this study, the needle served to exert mechanical force to disrupt the hydrogen bond network at the interface between the semi-liquid metal oxide layer and the flexible substrate, thereby inducing detachment of the semi-liquid metal from the substrate. Moreover, ethanol exhibited exceptional wettability toward both the semi-liquid metal oxide film and the flexible substrate, enabling rapid coverage of the substrate and detached semi-liquid metal. This phenomenon inherently prevented re-adhesion of the semi-liquid metal to the substrate. Liquid metals exhibit high surface tension (~500–700 mN/m at 25 °C), driving droplets to spontaneously form a spherical shape. In air, the oxide film on liquid metal droplets adheres to the glass substrate, flattening the droplets and reducing their contact angle. In contrast, in ethanol solution, the substrate is wetted and covered by ethanol, which prevents the droplet oxide film from

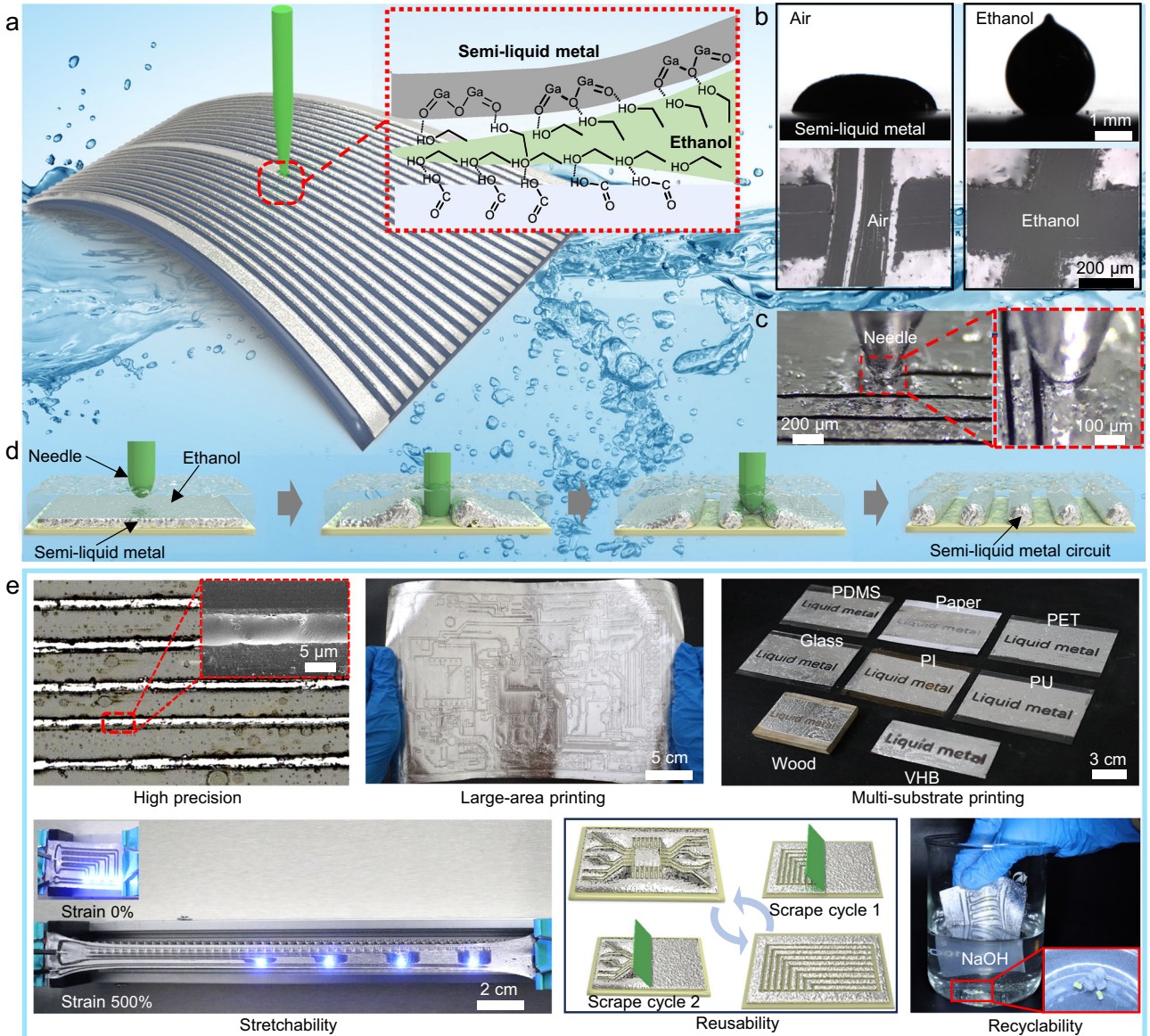

**Fig. 1 | Principles and advantages of the NECP method. a** Functional schematic of the NECP method. **b** Contact angles and cross-shaped scratch characterization of semi-liquid metal in ethanol and air. **c** Microscopic visualization of semi-liquid metal wire arrays fabricated via NECP method. **d** Mechanism of NECP method for semi-liquid metal circuits. **e** Key advantages of the NECP method.

contacting the substrate. Consequently, droplets are unaffected by substrate adhesion (no adhesive deformation), and their shape is dominated by surface tension, remaining spherical (Supplementary Fig. 6). Thus, semi-liquid metal immersed in ethanol showed a significantly larger substrate contact angle than that in air, indicating a non-adhesive state (Fig. 1b). Furthermore, the ethanol filled cross channel acted as a physical barrier, effectively preventing re-adhesion of the semi-liquid metal on either side to the original scratch location. Figure 1c shows the microscopic image of needle-tip non-loss-engraved semi-liquid metal wire arrays and the video is available in Supplementary Video 1. The arithmetic mean roughness (Ra = 5.22 μm) of the semi-liquid metal film was calculated based on the measured contour curve by the laser confocal microscope, as shown in Supplementary Fig. 7. As illustrated in Fig. 1d, when a substrate uniformly coated with semi-liquid metal is immersed in an ethanol environment and the needle tip traverses the substrate, ethanol reduces the adhesion between the liquid metal and the substrate, thereby preventing the needle tip from adhering to the liquid metal. Consequently, the mechanical force exerted by the needle tip propels the liquid metal to

both sides of the channel, increasing the thickness of the liquid metal layers on either side. Simultaneously, the exposed substrate is rapidly wetted by ethanol, which served to inhibit backflow of the semi-liquid metal. A single wire is formed by two parallel channels, whereas multiple parallel channels yielded a liquid metal wire array with no discernible loss of semi-liquid metal throughout the process. Through precise control of the needle tip trajectory, the NECP method enables the fabrication of complex liquid metal circuit patterns, as demonstrated in the operational flowchart of Supplementary Fig. 8. Figure 1e graphically illustrates the multifaceted advantages of this method, encompassing low cost, simple and efficient process, high precision, large-area printing capability, multi-substrate compatibility, high stretchability, reusability, and recyclability. Notably, this method enables the fabrication of wire arrays with a resolution as low as 5 μm. Owing to its avoidance of harsh processing steps and expensive equipment, the approach exhibits low cost, accommodates circuit fabrication across multiple scales ranging from micrometers to decimeters, and demonstrates compatibility with diverse substrates, including VHB tape, polydimethylsiloxane (PDMS), polyethylene

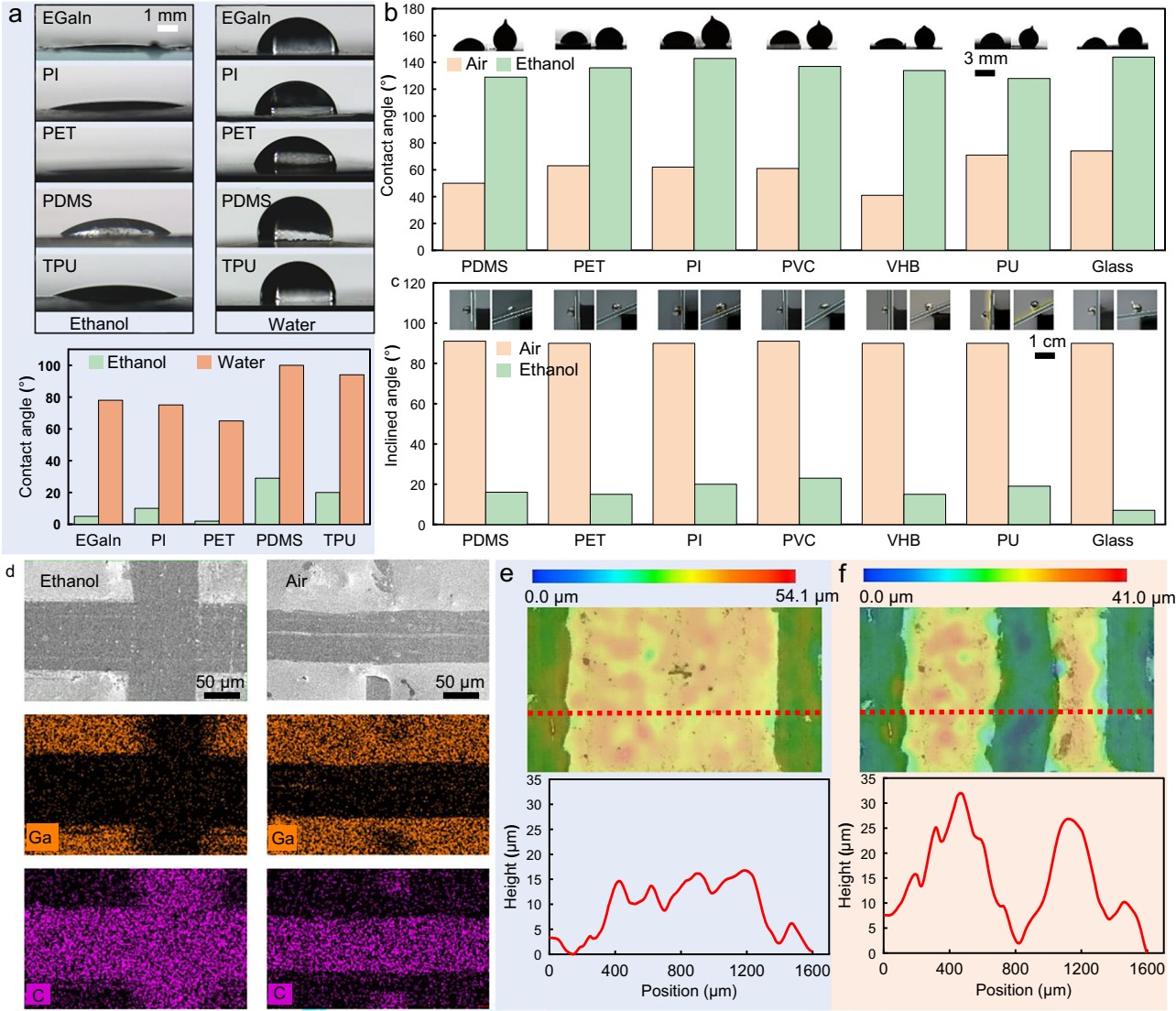

**Fig. 2 | Mechanisms for regulating interfacial adhesion between semi-liquid metal and substrates. a** Contact angles of ethanol and water on multiple substrates. **b** Contact angles of liquid metal droplets with various substrates in air and ethanol. **c** Inclined plane experiments of liquid metal droplets in air and ethanol. **d** Gallium element distribution maps at cross-shaped scratches in air and ethanol. **e** Microstructural morphology of semi-liquid metal before scratch formation. **f** Microstructural morphology of semi-liquid metal after scratch formation.

terephthalate (PET), polyimide (PI), polyurethane (PU), wood, glass, and paper. This versatility enables the technology highly adaptable for manufacturing a wide range of flexible electronic devices. Critically, given the scarcity of semi-liquid metal materials, the method achieves nearly complete recycling of semi-liquid metal. With no material loss incurred, this method facilitates circuit reusability, thereby significantly reducing technological application costs and mitigating both resource waste and environmental pollution. Supplementary Fig. 9 illustrates the needle tip moving trajectories during the fabrication of large-area, stretchable, multi-substrate and recyclable circuits. To systematically characterize the technological breakthroughs relative to traditional methods, the study established a comparative framework across six key dimensions: manufacturing cost, recyclability, conductivity loss, material loss, resolution, and printability on diverse substrates, with detailed data presented in Supplementary Table S1.

### Mechanisms for regulating interfacial adhesion between semi-liquid metal and substrates

The NECP method employed ethanol to suppress re-adhesion of liquid metal to the substrate, thereby enabling high-precision fabrication of

circuit patterns. Thus, we systematically investigated the wetting behavior of ethanol on semi-liquid metal and flexible substrate surfaces during the engraving of flexible liquid metal circuits, along with the resultant regulatory mechanisms governing interfacial adhesion. The study began by assessing the wettability of two liquids (ethanol and water) on multiple substrates via contact angle measurement experiments, as depicted in Fig. 2a and Supplementary Video 2. In accordance with the theory of contact angle and wettability, a smaller contact angle signifies superior wettability. The results reveal that ethanol exhibited contact angles below 30° on diverse substrates, including liquid metal, PI, PDMS, PET, and Thermoplastic Polyurethane (TPU), thereby demonstrating high wettability. Conversely, water showed substantially larger contact angles (all exceeding 65°) on these substrates, indicative of reduced wettability. Notably, water droplets on PDMS and TPU substrates displayed contact angles exceeding 90°, classifying these interfaces as hydrophobic. Water molecules form strong hydrogen bonds, leading to high surface tension (-72 mN/m at 20 °C) and poor spreading on substrates. In contrast, ethanol molecules have fewer and weaker hydrogen bonds, resulting in much lower surface tension than water (-22.3 mN/m at 20 °C), that enables ethanol

to spread more easily on substrates. Additionally, water only forms strong adhesion on polar substrates, while ethanol (with both polar and non-polar molecular characteristics) achieves strong adhesion on both polar and non-polar substrates, giving it broader applicability and more stable wetting performance. Thus, compared to water, ethanol exhibited superior wettability toward liquid metal and diverse flexible substrates, enabling efficient spreading and wetting on the surfaces of semi-liquid metal and flexible substrates during non-loss engraved patterning. Additionally, contact angles of liquid metal droplets on multiple substrates were measured in air and ethanol environments, respectively, as shown in Fig. 2b. The data discloses that liquid metal droplets exhibit smaller contact angles on all substrates in air than in ethanol. For instance, the contact angle of liquid metal droplets with the on PDMS substrate was approximately 50° in air, whereas it increased to approximately 130° in ethanol. The same trend was observed for PET, PI, Polyvinyl Chloride (PVC), VHB, PU, and glass. In air, the liquid metal droplet showed contact angles below 90° with all substrates, indicative of strong interfacial adhesion. In contrast, contact angles were significantly above 90° in ethanol, reflecting weak interfacial adhesion. Furthermore, liquid metal droplets in air and ethanol environments were positioned on inclined planes fabricated from the seven aforementioned substrates. As depicted in Fig. 2c, liquid metal droplets exhibited robust adhesion to these substrates, remaining stably attached even when the substrate was tilted to 90°. Conversely, ethanol -covered droplets failed to adhere stably. For instance, on a glass substrate inclined at just 7°, the droplet slipped. Results from contact angle and inclined-plane experiments indicate that ethanol spreading on semi-liquid metal and flexible substrate surfaces isolates the two materials, disrupting hydrogen bond interactions between the oxide film on the semi-liquid metal surface and the substrate. This disruption prevents re-adhesion of the semi-liquid metal to the flexible substrate, as schematically shown in Supplementary Fig. 10.

Second, this study employed a needle tip to exert mechanical pressure on semi-liquid metal, inducing its detachment from the substrate. Unlike the peeling of solid metal from the substrate, semi-liquid metal undergoes positional displacement due to its fluidity under mechanical loading. Although this process involves no metal loss, the extruded semi-liquid metal coalesces with the semi-liquid metal on both sides, potentially impacting the electrical characteristics of the circuit. Therefore, we investigated the flow dynamics of semi-liquid metal during needle tip extrusion. High-speed cameras were utilized to record the process of forming cross-shaped scratches by applying pressure with a metal needle tip to semi-liquid metal coatings in air and ethanol environments, respectively, as documented in Supplementary Fig. 11. In air, adjusting the moving speed of the needle tip cannot eliminate the semi-liquid metal adhesion at the cross, as shown in Supplementary Fig. 12. In air, when the metal needle tip engraves the semi-liquid metal coating, most of the semi-liquid metal is displaced from its original position. Notably, as the needle tip traverses the cross-intersection, semi-liquid metal from both sides is forced into the original scratch and reattaches to the substrate, obscuring the initial trace. In contrast, when in ethanol, the original scratch is infiltrated with ethanol, and even as the needle tip traverses the cross-intersection, semi-liquid metal from both sides remains unable to be forced into the original scratch, thereby precluding coverage of the initial trace. Additionally, gallium element distribution maps at cross-shaped scratches in air and ethanol environments (Fig. 2d) reveal that in air, the needle tip drives semi-liquid metal from both sides into the original scratch, causing circuit failure, whereas ethanol mitigates this issue. More critically, the persistence of minute semi-liquid metal residues in scratches during air-based engraving poses a risk of accidental circuit short-circuits, whereas engraving in ethanol fully displaces the metal from scratches, effectively precluding such short-circuits. This discrepancy is attributed to the adhesion of liquid metal

to both the needle tip and the substrate in air, which leads to residual liquid metal along the needle tip's path and incomplete material isolation. In ethanol, the needle tip and scratch are both wetted by ethanol, which acts as a physical barrier to liquid metal, thereby leaving no residue in the scratch. Furthermore, the high adhesive force between the semi-liquid metal and the substrate in air can overcome the cohesive force induced by the semi-liquid metal's high surface tension, enabling the semi-liquid metal to adhere to the scratch. In contrast, in ethanol, the needle tip, the semi-liquid metal, and the substrate are isolated from one another. As a result, the semi-liquid metal cannot adhere to the substrate; its cohesive force prevents it from being pushed out by the needle tip and also keeps it from adhering to the scratch, as shown in Supplementary Fig. 13. Finally, we characterized the microstructural evolution of the semi-liquid metal flanking the scratch before and after engraving, as depicted in Fig. 2e, f. Through contour curve analysis across Fig. 2e, f, we observed a marked post-engraving increase in semi-liquid metal thickness on both scratch flanks, evidence of metal extrusion from the scratch into lateral regions. This process enhances the cross-sectional area of lateral wires, thereby boosting conductivity, while achieving material-loss-free operation.

## Electrical characterization of semi-liquid metal wires via NECP method

The schematic illustration of the self-assembled 2-axis moving platform employed for the NECP method is depicted in Fig. 3a. A glass plate uniformly coated with semi-liquid metal was positioned on the 2-axis moving platform and immersed in ethanol. The 2-axis moving platform precisely manipulated the tip to etch the substrate surface, creating scratches with defined geometric configurations and achieving patterning (Supplementary Video 3). The NECP method of semi-liquid metal showcases a high level of precision and flexibility, enabling the fabrication of high-precision circuits with intricate patterns, as illustrated in Fig. 3b. Supplementary Fig. 14 reveals the movement path of the metal needle on the glass plate. Supplementary Fig. 15 presents more detailed information about the semi-liquid metal circuits on the glass plate, encompassing linear and circular arrays. Furthermore, the NECP method is suitable for fabricating circuit patterns with repeatability and high density, such as wireless charging coils (Supplementary Fig. 16). Herein, the width of the scratches can be adjusted by changing the width of the needle tip, thereby enabling the fabrication of coils with different spacing (Fig. 3c). Using a thinner needle tip can significantly reduce the coil spacing to minimize the coil size. Additionally, the needle tip squeezes the semi-liquid metal at the scratch into the wires on both sides, leading to an increase in the height of the wire edges, which can reduce the resistance of the wires, as shown in Supplementary Fig. 17. To verify the ability of this method to precisely control the width of the wires, wires with designed widths of 10 μm, 50 μm, 100 μm, and 200 μm were fabricated. The 3D contour diagrams and height information are presented in Fig. 3d. The results indicate that the cross-section of the semi-liquid metal wires is arched, and as the wire width increases, both the height and cross-sectional area of the wires increase substantially. This is due to the arc-shaped metal needle tip causing width variations in the scratch across different height levels. Additionally, the semi-liquid metal, exhibiting fluidity, tends to contract toward the scratch edges due to surface tension, resulting in an inverted trapezoidal cross-section (Supplementary Fig. 18). For closely spaced scratches, the arc tip displaces semi-liquid metal in higher regions, significantly reducing their height. Conversely, with larger spacing, the tip does not displace the metal in higher regions, allowing the wire's middle section to retain the initial coating height. Owing to the uneven distribution of solid metal particles within the semi-liquid metal and the non-uniform substrate wettability induced by ethanol, the wire cross-sections exhibit significant irregularities. For instance, different segments of the same wire may appear

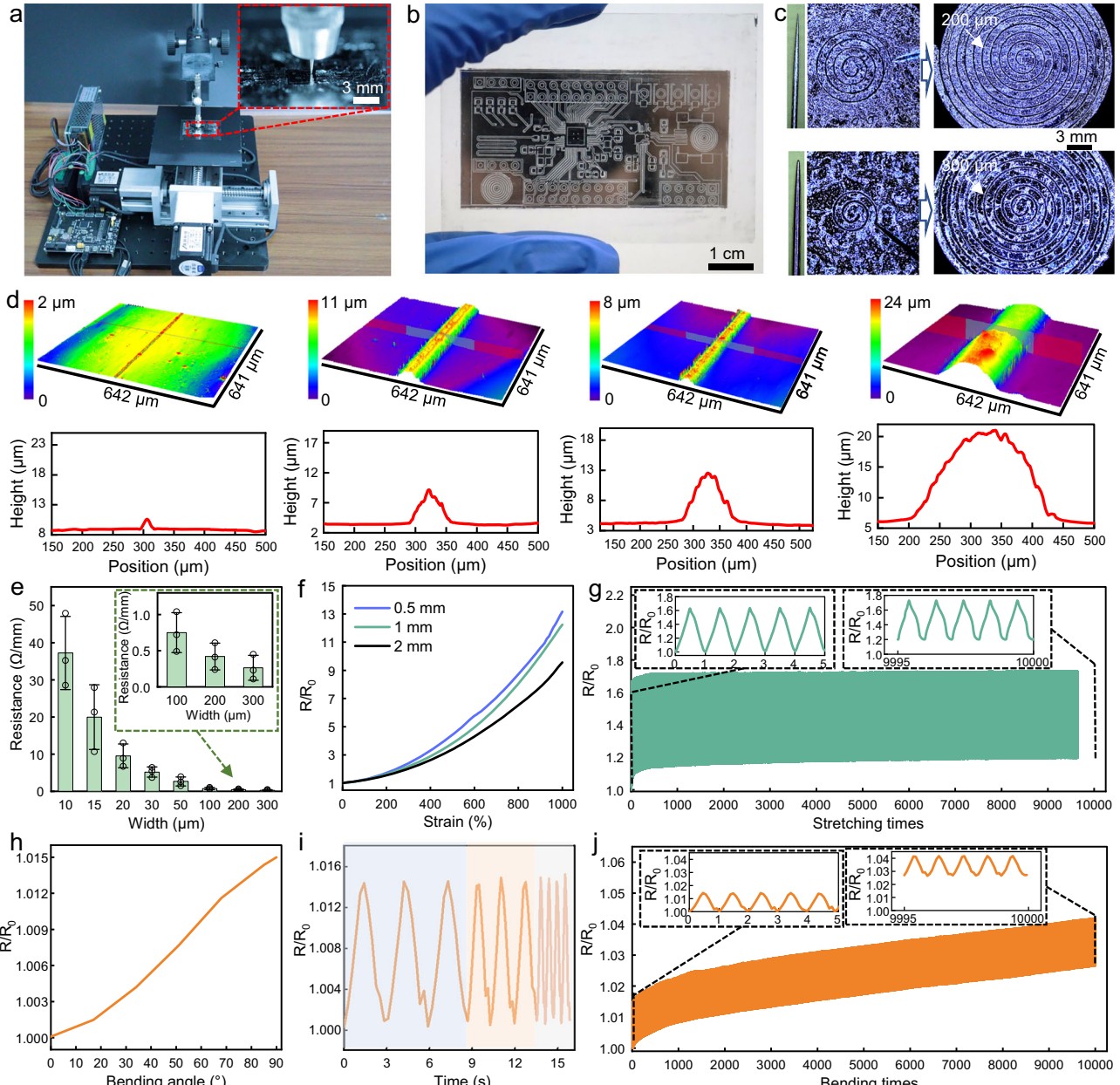

**Fig. 3 | Electrical characterization of semi-liquid metal wires via NECP method.**
**a** Photograph of the self-assembled NECP liquid metal two-dimensional mobile platform device. **b** Complex high-precision circuits fabricated via the NECP method. **c** Two different spaced coils made by two different diameters of the tips. **d** 3D contour plots and height information of semi-liquid metal wires with different widths. **e** Electrical resistance of semi-liquid metal wires with different widths (Data

are represented as mean ± s.d. $n = 3$ samples). **f** Resistance variation of the three types of wires under different stretching states. **g** Resistance variation during 10,000 cyclic stretching cycles. **h** Relationship between resistance variation and bending angle during semi-liquid metal wire bending. **i** Resistance variation when stretching liquid metal wires at different rates. **j** Resistance variation during 10,000 cyclic bending cycles.

flattened, elliptical, or locally bulged, as illustrated in the cross-sectional micrographs of Fig. 3d. This irregularity renders the cross-sectional area impossible to define in a stable and accurate manner. Consequently, calculating electrical conductivity via the conventional formula becomes unfeasible. To address this, we provided resistances for semi-liquid metal wires of different widths (Fig. 3e), and it is evident that the resistance decreases significantly as the wire width increases. From the microscopic images of semi-liquid metal wires with varying widths shown in Supplementary Fig. 19, it can be observed that there are slight defects at the edges of each wire. When the overall width of the wire is small, these defects can lead to significant changes in its resistance. Conversely, when the overall width of the wire is large, the impact of these defects gradually diminishes. For the wire with a width

of 10 μm, its resistance value can reach 38.2 Ω/mm, and the resistance difference of the samples ranges from 28.5 Ω/mm to 47.9 Ω/mm, exhibiting a relatively large error. In contrast, when the wire width exceeds 100 μm, the resistance difference of the samples can be reduced to within 0.5 Ω/mm, and the consistency of the resistance value is excellent, which can meet most of the requirements of flexible electronics.

Subsequently, semi-liquid metal wires on VHB substrate and PI substrate (width: 1 mm, length: 2 cm) were respectively subjected to stretching and bending tests. Figure 3f demonstrates the relationship between the resistance variation and the stretching rate during the stretching process. Benefiting from the high stretchability of the VHB substrate and the fluidity of the semi-liquid metal, the semi-liquid

metal can follow the stretching of the substrate up to 1000%, and the wire remains connected throughout the entire process. The bulk semi-liquid metal with initial resistance of 1.65 Ω shows significant resistance differences depending on the change of geometric shape, and its resistance variation at a stretching rate of 1000% can reach 12.2, indicating its potential application as a strain sensor. The semi-liquid metal wires exhibit reliable and stable responses under repeated application of strain (more than 10,000 cycles), and no electrical failures occur (Fig. 3g). The semi-liquid metal wires possess excellent overall stability during the cyclic stretching process, and the resistivity shows periodic fluctuations at different stages of the stretching process, with good repeatability of the resistivity change during each stretching-recovery process, suggesting a good mechanical-electrical response repeatability. After 10,000 stretching cycles, the increase in the resistance of the wire was attributed to the creep of the VHB substrate, which prevented it from returning to its initial length. Additionally, when the semi-liquid metal wire was bent by 90° along the PI substrate, its resistance variation can reach 1.015 (Fig. 3h). Figure 3i shows that when the wire was stretched at different states, the resistivity demonstrated good periodic repetition, indicating that the wire is sensitive to the stretching rate and can adapt to various mechanical environments. Figure 3j illustrates the resistance variation during 10,000 bending cycles. In the early cyclic bending of the substrate, part of the liquid metal in the semi-liquid metal flows to both sides under gravity and base vibration, reducing the cross-sectional area at the middle bending point and increasing resistance. However, as bending cycles increase, the liquid metal gradually stops flowing. Then, the slight rise in the s resistance of semi-liquid metal is mainly due to surface oxidation. Although there is an overall slow upward trend, the resistance variation is only 1.04, and the resistance variation during the bending-recovery process has good repeatability, demonstrating the stability and repeatability of the mechanical-electrical response. Furthermore, the wire sample underwent 10,000 cyclic stretches at a higher strain (500%). As shown in Supplementary Fig. 20, the wire remained conductive after 10,000 stretches under high strain, while its resistance variation increased to 5.83. These results align with previous reports, attributed to gradual relaxation of the stretchable substrate (failing to recover initial length) after repeated stretches, combined with continuous oxide formation on the semi-liquid metal during cyclic stretching, which both leading to progressive resistance increase. Thus, this semi-liquid metal wire is suitable for stretchable conductors in flexible circuits but not for direct use as strain sensors, as resistance correction is required after multiple stretches. In addition, thinner (width: 0.5 mm) and thicker (width: 2 mm) semi-liquid metal wires were tested for stretching (Fig. 3f) and bending (Supplementary Fig. 21). Results showed the resistance variation of 2 mm-wide wire with initial resistance of 0.78 Ω reached 9.5 at maximum stretch, and it withstood 10,000 cycles of 200% stretching without fracture. Similarly, the resistance variation of 0.5 mm-wide wire with initial resistance of 4.25 Ω reached 13.2 at maximum stretch, and also endured 10,000 cycles of 200% stretching without breaking. For bending performance, the 2 mm-wide wire exhibited a resistance variation of only 1.004 at 90° bending and 1.04 after 10,000 bending cycles. Similarly, the 0.5 mm-wide wire showed 1.012 (90° bending) and 1.06 (10,000 cycles), respectively. Additionally, long-term property tests were performed on the semi-liquid metal. The resistance of 1 mm-wide, 2 cm-long semi-liquid metal wires and their resistance variation were measured over a two-month storage period (Supplementary Fig. 22). Results show the resistance variation of unencapsulated wires increased to 1.14, while that of encapsulated ones only reached 1.01. Moreover, after long-term storage, the wires still achieved large tensile deformation with the substrate while maintaining electrical continuity. The resistance variation at maximum stretch was 12.5, comparable to that of the original wires. These results confirm the long-term electrical and mechanical stability of semi-liquid metal. Finally, to confirm its

integration capability with traditional electronic devices, we used HCl solution to deoxidize the pins of traditional electronic components[44], such as LEDs, enabling them to be wetted and soldered with semi-liquid metal wires, forming stable electrical interconnections, as shown in Supplementary Fig. 23. Additionally, a rigid film (polyethylene) was embedded into the VHB tape to minimize deformation at the connection, as shown in Supplementary Fig. 24a. Supplementary Fig. 24b, c illustrate that during the 200% stretching process, the voltage and current across the LEDs gradually decreased, which is attributed to an increase of 0.8 Ω in the resistance of the semi-liquid metal wire. However, the increased wire resistance only caused a voltage drop of 4 mV and a current reduction of 4 μA across the LEDs. Therefore, the variation in wire resistance induced by deformation has a negligible impact on the circuit function. This stable wetting and soldering structure with rigid electronic components ensure the subsequent development of multifunctional flexible electronic devices.

## Characterization of non-loss, recyclability, reusability, and multiple substrate-applicability to for NECP method

The NECP method enabled high-precision manufacturing of complex circuits using a 2-axis moving platform. Figure 4a demonstrated a semi-liquid metal wire array with linewidth of 50 μm. The array was attached to a PI film, which was conformally coated onto non-planar surface (fingernail tip), due to its thin thickness (50 μm). This feature makes it highly suitable for applications in skin-integrated bioelectronic devices. In addition, some electronic devices require transparency, such as optoelectronic devices fabricated on transparent substrates like PDMS. Therefore, in addition to engraving the semi-liquid metal circuit pattern, excess semi-liquid metal can be precisely directed to desired regions by modifying the trajectory of the metal needle, as exemplified by the LED array manufactured on a PDMS substrate in Fig. 4a. The needle's movement path is provided in Supplementary Fig. 25. Additionally, mesh metal patterns with varied widths and pitches were fabricated (Supplementary Fig. 26). The mesh wires have widths of 1 mm and 0.5 mm, with corresponding spacings of 1 mm and 0.5 mm. All wires were produced via the second engraving method, where semi-liquid metal in the mesh holes is fully pressed into the wires. This engraving mode can not only minimize the coverage area of the semi-liquid metal, but more importantly, the pushed semi-liquid metal can keep the resistance value of the wire stable and improve the uniformity of the wire. As shown in Fig. 4b, a copper sheet and a semi-liquid metal wire with an initial width of 1 cm were successively cut to 1 mm, with a reduction of 1 mm each time. Since the semi-liquid metal is pushed into the adjacent wire by the needle tip, the overall volume of the wire remains almost unchanged. According to the conductor resistance calculation formula:

$$R = \frac{\rho L}{A} = \frac{\rho L}{V/L} = \frac{\rho L^2}{V}$$

where $L$ is the wire length, $R$ is the resistance, $\rho$ is the resistivity, $V$ is the volume, and $A$ is the cross-sectional area. The resistance should theoretically remain unchanged. Furthermore, after each engraving step, the noticeable increase in wire edge thickness also confirms the constant volume of wire (Supplementary Fig. 27). Experimental results highlight the distinct electrical behavior of copper strips and semi-liquid metal wires during dimensional reduction. When reduced from an initial width of 1 cm to 1 mm, copper resistance increased by tenfold, consistent with expected resistivity variations based on width changes. In contrast, semi-liquid metal wires exhibited stable or even slightly decreased resistance when reduced from 10 mm to 5 mm, and a gradual increase to 115% of the initial value as the width decreases further to 1 mm. Observations from pictures in Fig. 4b and Supplementary Video 4 revealed semi-liquid metal thickness increased during the narrowing process, which leading to more uniform distribution

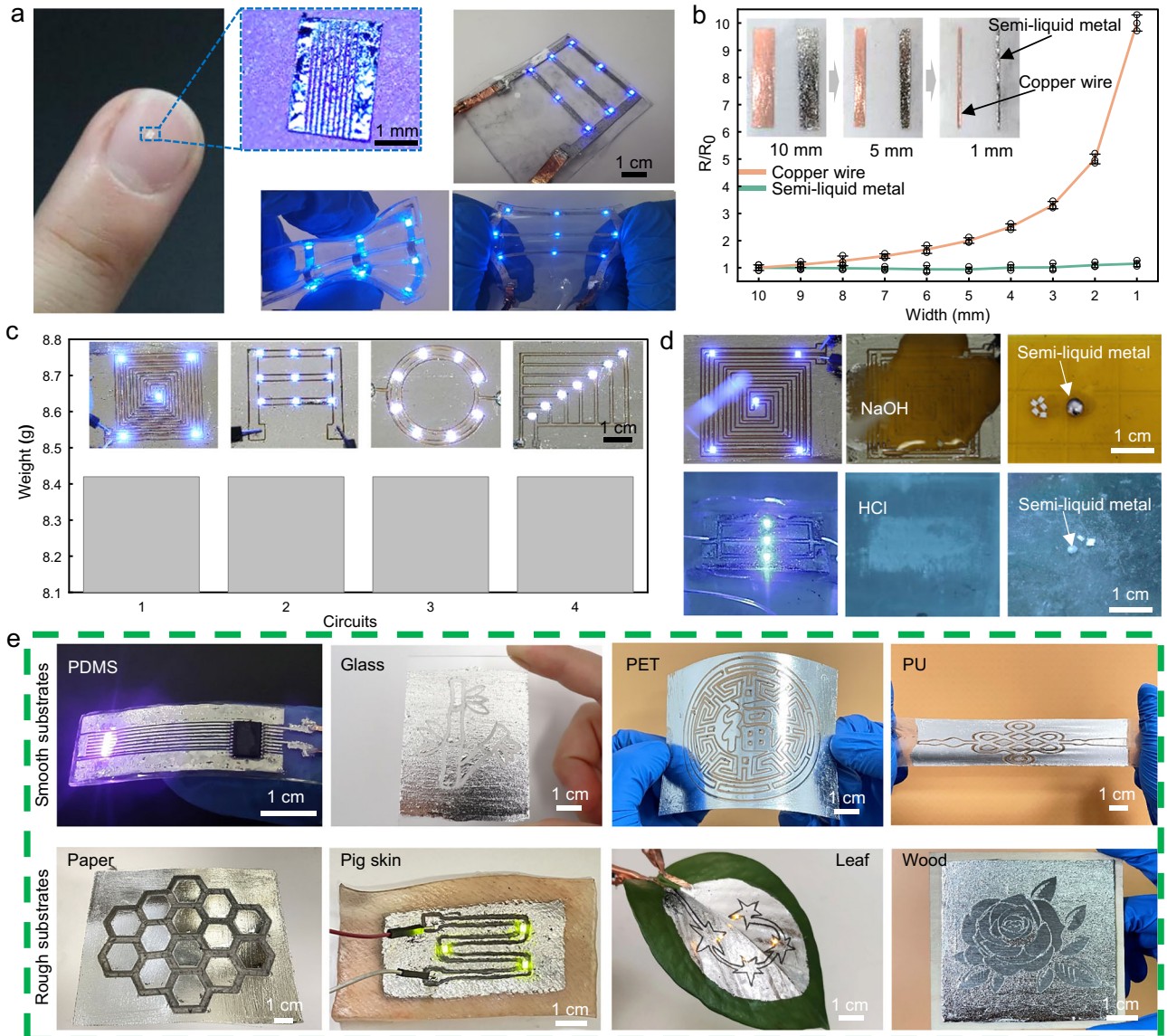

**Fig. 4 | Characterization of non-loss, recyclability, reusability, and multiple substrate-applicability to for NECP method. a** Semi-liquid metal wire array with linewidth of 50 μm and transparent LED circuits on PDMS. **b** Resistance variation of a copper wire and semi-liquid metal as the line width was reduced from 1 cm to 1 mm (Data are represented as mean ± s.d. n = 3 samples). **c** Successive fabrication of four distinct circuits on the same glass substrate, where each preceding circuit was scraped flat prior to the next fabrication. **d** Recovery process of semi-liquid metal from LED arrays in NaOH and HCl solutions. **e** The non-loss engraved semi-liquid metal patterns on eight different substrates.

and lower resistance. However, as a drawback of the second engraving approach, prolonged reduction caused irregularities at the wire edges due to ethanol effects, resulting in a slight increase in resistance. Therefore, the second engraving approach is not suitable for manufacturing high-precision wires.

Moreover, the NECP method enabled highly reconfigurable circuit fabrication without material loss, as demonstrated by its ability to erase and reuse patterns. As illustrated in Fig. 4c, four distinct circuits were successfully created on the same glass substrate. After each circuit was fabricated, it could be erased using a simple scraping technique, enabling the creation of subsequent circuits on the same base. The fractures caused by the metal needle can be completely filled with semi-liquid metal using a scraper, restoring the resistance to the state before the pattern was engraved. even when the width of the fracture reaches 5 mm, as shown in Supplementary Fig. 28. The needle's movement path for these circuits is provided in Supplementary Fig. 29. Remarkably, the mass of semi-liquid metal remained consistent at

8.42 g after each sequential fabrication process. In addition, a large number of repeated engraving operations were carried out to evaluate the material loss caused by multiple repeated usages. The results shown in Supplementary Fig. 30 indicate that during the engraving process repeated 50 times, the mass of the semi-liquid metal remains at 8.42 grams throughout. These results underscore the exceptional reusability and resource efficiency of this method, eliminating material waste in electronic manufacturing. The arbitrary disposal or improper handling of liquid metal may lead to environmental contamination, as it can enter soil and water systems. Consequently, achieving effective recycling of liquid metal is crucial for reducing its impact on the environment and preventing resource waste while cutting down costs. Here, as a proof-of-concept, we demonstrated the recovery process of semi-liquid metal from LED arrays in NaOH solution (1 mol/L) and HCl solution (1 mol/L), as shown in Fig. 4d. When NaOH solution was applied to the PI substrate with semi-liquid metal circuits, the semi-liquid metal layer separated from the PI substrate instantly and

eventually formed a single droplet (Supplementary Video 5). This occurs because $Ga_2O_3$ is an amphoteric oxide, which reacts with NaOH according to the following chemical equation:

$$Ga_2O_3 + 2NaOH = 2NaGaO_2 + H_2O$$

In this reaction, $Ga_2O_3$ reacts with NaOH to form sodium metagalate ($NaGaO_2$) and water. Therefore, the semi-liquid metal loses its adhesion to the substrate and aggregates into droplets under high surface tension. Furthermore, we fabricated a semi-liquid metal-based LED array on a PVA substrate. When immersed in HCl solution, the PVA film dissolved, causing the LED to extinguish instantly and the semi-liquid metal to separate from the PVA membrane, ultimately yielding a single semi-liquid metal droplet and LED components (Supplementary Video 5). Unlike the NaOH solution, when $Ga_2O_3$ reacts with HCl, it forms gallium chloride ($GaCl_3$) and water:

$$Ga_2O_3 + 6HCl = 2GaCl_3 + 3H_2O$$

Thus, the semi-liquid metal separated from the PVA membrane in HCl and eventually aggregates into droplets. The dissolution process completed within approximately 300 s at 60 °C with a constant rate, allowing the dissolution time to be controlled by varying the thickness of the PVA film and the heating temperature. Finally, we compared the mass of recovered semi-liquid metal (Supplementary Fig. 31) before and after recycling in NaOH and HCl solutions. The results indicated that only 3.35% and 2.67% of material loss occurred in each recycling process, significantly reducing the resource waste and the raw material costs.

Prior studies have demonstrated that semi-liquid metal exhibits high adhesion to various smooth substrates due to hydrogen bonding interactions at the oxide-substrate interface, enabling uniform coating on these surfaces[45]. Besides, the solid metal particles within semi-liquid metal experience increased friction with rough substrates, allowing external pressure to force the material onto such surfaces[46]. Leveraging these properties, semi-liquid metal can be patterned across diverse substrates, opening possibilities for versatile circuit manufacturing. Therefore, NECP method was used to fabricate semi-liquid metal patterns on various substrates with different surface chemistries and roughness. As illustrated in Fig. 4e, the method was tested on eight types of substrates: four smooth substrates (PDMS, Glass, PET, PU) and four rough substrates (paper, pig skin, leaf, wood board). The needle's movement path for these substrates is provided in Supplementary Fig. 32. These substrates span a wide range from polymers to everyday items and biological tissues, encompassing both rigid and flexible materials. This universality of this method exhibits potential applications in wearable devices, medical health, and Internet of Things (IoT), facilitating flexible integration and functional expansion across diverse fields.

### Improved NECP method for 3D curved electronics

In addition to the fabrication of planar circuits, the NECP method is also applicable to the fabrication of 3D curved electronics. However, when the planar circuit fabrication method described previously is used to fabricate 3D curved electronics, the entire 3D object must be immersed in ethanol, and the 2-axis moving platform is unable to control the movement of the metal needle on the 3D curved surface. Moreover, the semi-liquid metal possesses a certain level of flow-ability, and the structural stability of the semi-liquid metal on the 3D curved surface still needs to be further optimized. To address these three issues, we have carried out a series of modifications to the planar circuit fabrication equipment and developed a NECP method for 3D curved electronics, as shown in Fig. 5a. First, an ethanol-dipped marker pen was employed instead of metal needles, as shown in Supplementary Fig. 33. The marker pen's tip was wetted with

ethanol and isolated the semi-liquid metal during movement on 3D curved surfaces by depositing ethanol into the grooves. Second, a 5-axis moving platform (Supplementary Fig. 34) was assembled to fix 3D objects and precisely control the marker pen across their surfaces. Finally, the doping ratio of solid metal particles in the semi-liquid metal was adjusted to 20%, resulting in the semi-liquid metal with lower fluidity. This semi-liquid metal was able to maintain its initial shape for up to 10 days when coated on the curved surface. In contrast, the pure liquid metal gathered at the bottom under the influence of gravity right after it was coated on the curved surface, as shown in Supplementary Fig. 35. This semi-liquid metal with lower fluidity was evenly applied to the 3D curved surface, and the NECP method of the semi-liquid metal on the 3D curved surface was accomplished using the ethanol pen and the 5-axis moving platform, as demonstrated in Supplementary Video 6. To demonstrate the method's versatility, some LED arrays were conformally applied to various 3D-printed models with complex shapes like spherical, wavy, and arc-shaped surfaces, as shown in Fig. 5b. This method also worked across diverse materials, including objects like pepper, egg, and mouse (Fig. 5c). Supplementary Fig. 36 illustrates the ethanol pen's movement paths on these surfaces.

The NECP method demonstrated remarkable operational simplicity and broad universality. The conformal 3D electronic devices fabricated by this method can perfectly conform to the surfaces of 3D curved objects, and they possess extensive application prospects in various fields such as aerospace and smart home. For instance, aircraft icing in cold weather poses risks by increasing weight and energy consumption. Traditional de-icing methods, such as spraying chemical de-icing fluids on the wings or installing hot air de-icing systems, are inefficient and costly. In contrast, this method enabled conformal anti-icing systems on aircraft surfaces, as shown in Fig. 5d. This de-icing system was composed of an electrically heated wire and an LED array. Supplementary Fig. 37 shows the movement path of the ethanol pen on the curved surface of the fuselage. As shown in Fig. 5e, the electrically heated wires can effectively heat the wing under a current of 3 A, increasing the surface temperature of the wing from 23.5 °C to 60.7 °C within 80 s. The pattern of the LED array was customizable and could provide guidance for night flights. The part of the LED array also can be replaced by other functional circuits to meet different requirements. In addition, conformal electronics demonstrate significant application value in the field of smart home. They can closely adhere to the complex shapes of household items to achieve the integration of electronic circuits. While maintaining the aesthetic harmony of the equipment, they endow households with functions of intelligent monitoring and control, greatly enhancing the intelligence and convenience of the home. Figure 5f shows the LED array circuits implemented on a house model using the NECP method, covering substrates such as the glass roof windows, wooden tables, and plastic floors. Supplementary Fig. 38 shows the movement paths of the ethanol pen on the surfaces of three parts of the house model. The above results indicated that the NECP method enables precise fabrication of flexible circuits on diverse material surfaces without requiring complex pretreatment, offering operational simplicity and high compatibility.

### Demonstration of customizable and zero-waste skin-interfaced electrodes

The NECP method exhibits exceptional customizability, which significantly facilitates the fabrication of wearable devices. By tailoring designs to specific body areas or individual needs with unique dimensions and configurations, comfort and signal detection accuracy can be notably enhanced. Here, utilizing non-loss engraved semi-liquid metal circuits for the fabrication of skin-interfaced electrodes, we demonstrated the effectiveness of this method in monitoring electrophysiological signals such as electrocardiogram (ECG) and

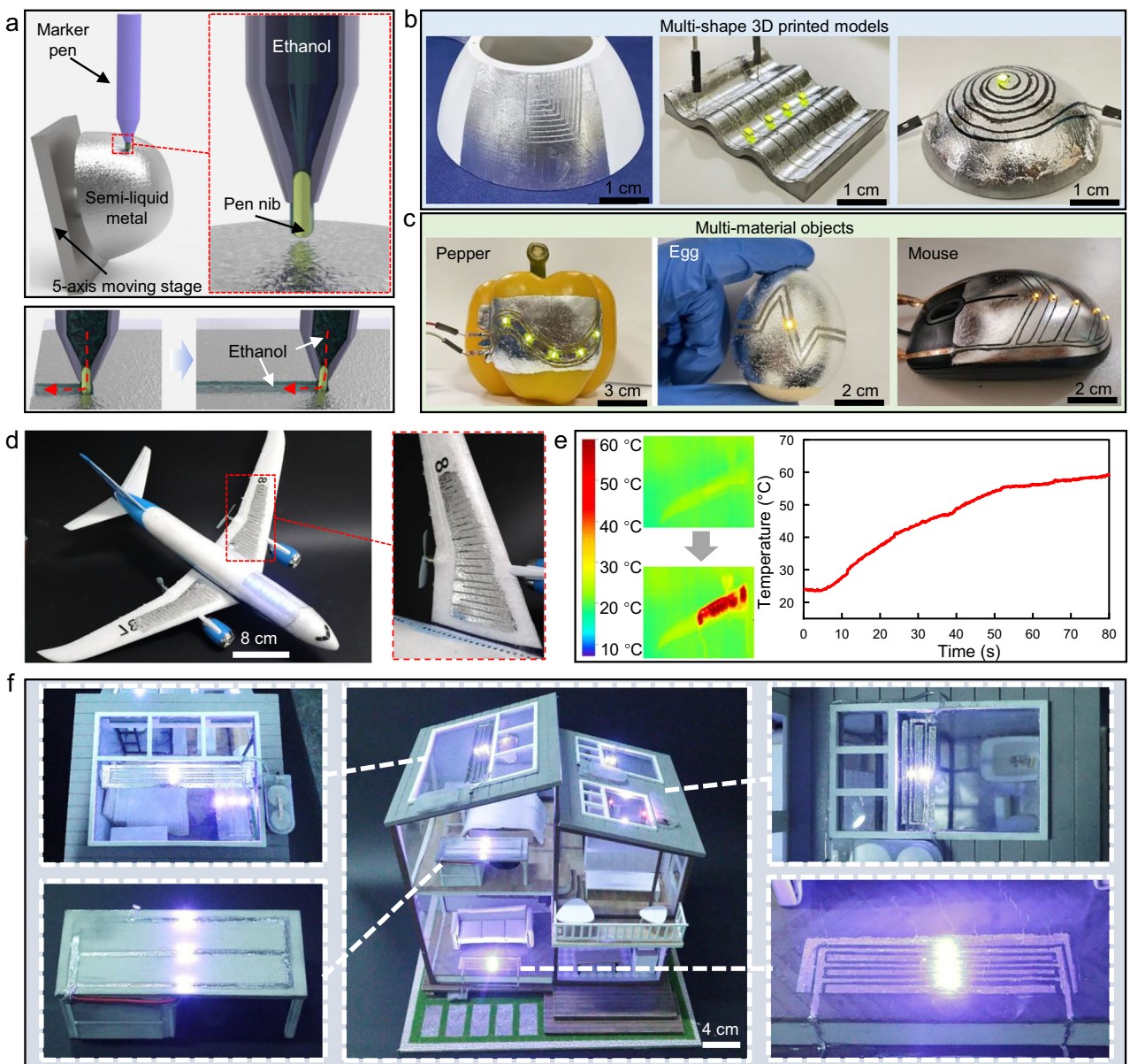

**Fig. 5 | Improved NECP method for 3D curved electronics. a** Schematic illustration of the non-loss engraved 3D pattern method. **b** LED arrays conformally applied to 3D models with diverse shapes. **c** LED arrays integrated onto multi-material objects. **d** Application of the NECP method in fabricating aircraft de-icing systems. **e** Infrared images and corresponding surface temperature of the wing. **f** LED array circuits fabricated using the non-loss engraved pattern method, implemented on a house model and covering different substrates.

electromyogram (EMG). As illustrated in Fig. 6a, the ECG monitoring system comprises semi-liquid metal electrodes printed on a PU film using the NECP method, along with a compact circuit board housed in a 3D-printed casing. The system was affixed to the chest area of a male volunteer via an adhesive film at the base of the PU film. Initially, semi-liquid metal was uniformly coated onto the PU film, electrodes were patterned using the NECP method subsequently (Fig. 6b). The collected signals were processed and transmitted via a microprocessor and Bluetooth module on the circuit board, enabling heart rate calculation at the computer end (Supplementary Fig. 39). In the experiment of human ECG signal acquisition, the ECG waveform was successfully obtained, and the P wave, QRS complex, and T wave could be clearly distinguished. The baseline drift and external interference were low. Enlarged waveforms revealed that the semi-liquid metal electrodes captured QRS complexes with comparable precision to traditional Ag/AgCl electrodes, with peak amplitudes and intervals within acceptable error ranges (Fig. 6c). This demonstrates the

potential of semi-liquid metal electrodes to match conventional medical electrodes in ECG signal quality. Over a 60-min monitoring period, the system remained stable, reflecting dynamic heart rate variations corresponding to physiological states, ranging from approximately 58 to 110 beats per min (Fig. 6d). Even during dynamic posture change such as lying, sitting, and standing, the collected ECGs exhibited intact characteristic waveforms with minimal baseline displacement (Supplementary Fig. 40), indicating the electrodes' ability to maintain conformal contact with skin during movement. The RR interval scatter plot displays spindle-shaped distribution, with data points clustering around the 45° line, suggesting sinusoidal rhythm regularity and normal cardiac function in the subject (Fig. 6e). These results underscore the capability of semi-liquid metal electrodes to precisely capture ECG features, offering an innovative solution for wearable ECG monitoring devices. Furthermore, Fig. 6f demonstrates the NECP method for the construction of the desired deformable EMG electrodes. The semi-liquid metal electrodes were attached to the male

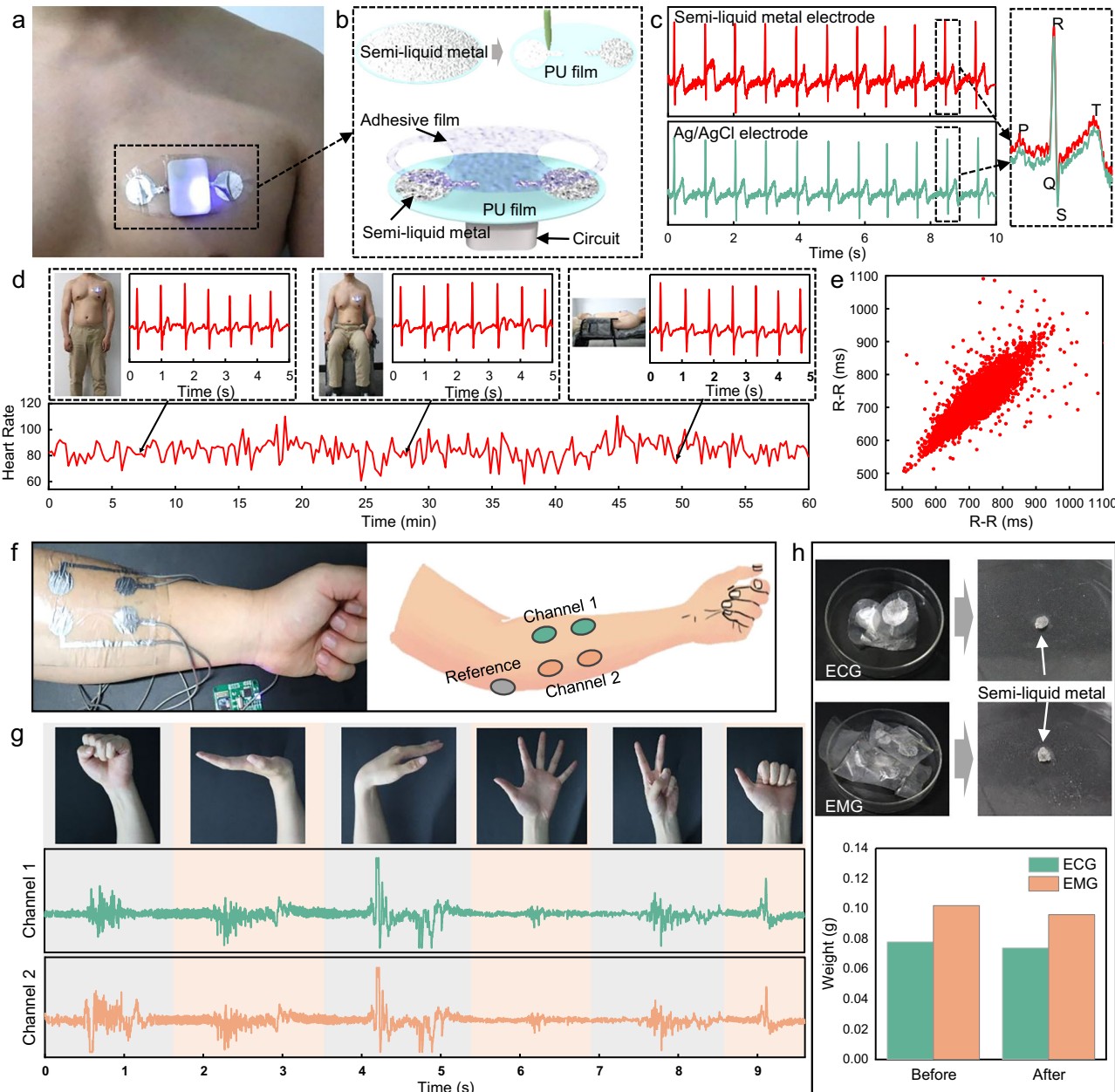

**Fig. 6 | Demonstration of customizable and zero-waste skin-interfaced electrodes. a** Photo of the wireless ECG monitoring devices. **b** Schematic diagram of the structure of the ECG monitoring device. **c** Comparison chart of ECG acquisition between the semi-liquid metal electrodes and commercial AgCl electrodes. **d** ECG waveforms and heart rate in standing, sitting, and lying positions. **e** Scatter plot of the R-R intervals. **f** Photo and channel schematic of the wireless EMG monitoring device. **g** Two channel EMG signals of 6 gestures. **h** Weight change of the ECG and EMG electrodes after recovery.

volunteer's forearm to construct a dual-channel EMG acquisition system (Supplementary Fig. 41). Benefiting from the fluidity and deformability of the semi-liquid metal, the electrodes can conformally adhere to the skin. Even when the skin is rubbed and stretched, the electrodes can stably adhere to the skin surface (Supplementary Fig. 42). Figure 6g shows the EMG signals of six typical hand gestures measured on the forearm using the semi-liquid metal electrodes, verifying its ability to sense high-fidelity EMG signals. Supplementary Fig. 43 presents the EMG signal-to-noise ratio (SNR, Channel 1) for the semi-liquid metal electrodes and traditional Ag/AgCl electrodes across 6 gestures. The semi-liquid metal electrodes exhibit higher SNR than Ag/AgCl electrodes under most gestures, and only in gesture 2 is their SNR slightly lower. Since higher SNR indicates less noise interference and better signal quality, the semi-liquid metal electrodes perform better in EMG signal acquisition for these 6 gestures. Finally, we

measured the impedance between the semi-liquid metal electrodes, traditional Ag/AgCl electrodes and the skin, and compared it with the impedance of the semi-liquid metal electrodes on moist skin, as shown in Supplementary Fig. 44. The semi-liquid metal electrodes have lower interfacial impedance than the traditional Ag/AgCl electrodes due to their excellent fluid deformability, which can adaptively fit the micro-topography of the skin surface, and higher electrical conductivity. When the skin sweats, the interfacial impedance between the semi-liquid metal electrode and the skin decreases. This is because sweat forms a continuous ionic conductive film on the skin surface. It not only fills the tiny gaps between the electrode and the skin, but also penetrates the stratum corneum to increase its water content, and enhance internal ion migration to reduce resistance. Additionally, sweat acts as a lubricant and filling medium, promoting the semi-liquid metal to fully wet the skin, lowering contact resistance and thus the

overall interfacial impedance. After the semi-liquid metal electrode is removed, slight benign residues may occasionally appear (similar to some commercial electrocardiogram gels). However, such residues can be easily and completely removed by simple wiping with an alcohol swab, which is a standard item in medical and personal care settings, as shown in Supplementary Fig. 45. Benefiting from the advantages of non-loss and recyclability of the NECP method, there was no loss of semi-liquid metal during the electrode manufacturing process, and NaOH solution was used to achieve nearly complete recycling of semi-liquid metal. Figure 6h displays that the ECG and EMG electrodes were detached from the substrate in the NaOH solution and merge into reusable bulk liquid metal droplets. The mass of the recovered semi-liquid metal droplets was almost consistent with that of the semi-liquid metal used for electrode fabrication. These demonstrations collectively indicate that the non-loss engraved semi-liquid metal circuits can conveniently and quickly provide customized and highly stable signal acquisition solutions for wearable physiological electrical signal monitoring devices. Such semi-liquid metal electrodes are expected to be applied to zero-waste medical electronic devices, addressing the environmental pollution and resource waste caused by disposable medical devices.

## Discussion

The NECP method developed in this study represents a transformative method in flexible electronics manufacturing, transcending the limitations of conventional additive/subtractive fabrication. By harnessing a dynamic interfacial adhesion regulation mechanism between liquid metal and substrates via ethanol, this technology achieves zero material loss during mechanical extrusion, thereby resolving the critical challenges of material waste, process complexity, and high operational costs inherent in traditional manufacturing approaches. Furthermore, the interfacial adhesion modulation strategy overcomes the issue of substrate specificity in liquid metal circuit fabrication caused by varying interfacial properties across different substrates. It enables the customization of flexible circuits with diverse performance characteristics, such as high stretchability and conformability, on a wide range of substrates. Additionally, the technology facilitates the recyclability and reusability of liquid metal. By eliminating metal waste through non-destructive etching, it mitigates environmental pollution and significantly reduces material costs, aligning with the principles of green manufacturing.

However, the current technical level makes mass production difficult to achieve, so this technology is only suitable for the rapid fabrication of personalized custom circuits. In our next step, we will optimize printing parameters in real time through machine learning algorithms, build fully automatic unmanned manufacturing equipment, and develop multi-scale needles ranging from micro-nano to macro scales. This will enable the entire manufacturing process from micron-level wires to centimeter-level wires to be completed on the same equipment, thereby improving manufacturing speed and realizing mass production. Looking ahead, this work is poised to accelerate the practical translation of liquid metal flexible circuits into cutting-edge applications such as wearable health monitoring systems and implantable diagnostic devices, thereby contributing to the advancement of environmentally sustainable and intelligent flexible electronic devices.

## Methods

### Materials

Gallium (Ga, 99.9%) and indium (In, 99.9%) were purchased from Anhui Minor New Materials Co., Ltd. Silver-plated copper microparticles (mean diameter: 15 μm, Ag content: 3 wt%) were purchased from Hebei Jingrui Alloy Products Co., Ltd. Anhydrous ethanol (99%) were were purchased from Xintai Yixinkang Medical Supplies Co., Ltd. Stainless-steel needles (NO.0517) were purchased from Dandong Shuangyan

Needle-making Co., Ltd. Sodium hydroxide were purchased from Xilong Scientific Co., Ltd.

### Fabrication of semi-liquid metal

Gallium with a mass fraction of 75.5% and indium with a mass fraction of 24.5% were mixed in a beaker, and the mixture was heated at 200 °C for 2 h to prepare liquid metal (EGaIn). Then, silver-plated copper microparticles with a mass fraction of 15% and 20% were added to the prepared EGaIn. Some sodium hydroxide aqueous solution (1 mol/L) was poured into the beaker. After stirring for 3 min, the silver-plated copper microparticles entered the liquid metal and semi-liquid metal was obtained.

### Fabrication of multiple substrates

PDMS (Sylgard 184, Dow Corning, Inc.) and Ecoflex (Beijing Tiantong Huayi Landscape Technology Co., Ltd.) were prepared using conventional methods. The VHB tape is a transparent universal acrylic adhesive tape (VHB™ 4905), was purchased from Minnesota Mining and Manufacturing Co., Ltd. Wood was purchased from Shuyang Shengteng Trade Co., Ltd. PI film (C81854-25MG) and PVA film was purchased from Sigma-Aldrich Corp. PU film was purchased from Zibo Tangjian Medical Devices Co., Ltd. Paper was purchased from Deli Group Co., Ltd. Glass was purchased from Nantong Bestest Experimental Equipment Co., Ltd.

Pattern devices. An assembled 2-dimensional moving platform was used to fabricate high-precision non-loss engraved semi-liquid metal circuits. A plotter (42HD00, Jinhua Jinqian Intelligent Technology Co., Ltd.) was utilized for creating complex and large non-loss engraved semi-liquid metal circuits. An assembled 5-axis moving platform was used to fabricate the non-loss engraved patterned semi-liquid metal circuits on the 3D curved surface. The ethanol pen was made by filling ethanol into an empty marker pen.

### Electrical characterization

The electrical conductivity of semi-liquid metal with varying doping ratios was determined via the standard four-point probe method. Prior to testing, the samples were placed and secured within a trapezoidal groove; this groove measured 50 mm in length and had a cross-sectional area of 15 mm². The resistances of semi-liquid metal wire samples with different widths were measured by a digital multimeter (Keithley 2002, Tektronix, Inc.). A four-terminal method was employed to avoid the influence of contact resistance. The semi-liquid metal wires (with a width of 1 mm and a length of 2 cm) on the VHB substrate and the PI substrate were fixed onto a dynamic mechanical test system (HC-01, Dongtai Suheng Transmission Technology Co., Ltd.), and a multimeter (U1251B, Agilent Technologies, Inc.) was used to measure the resistance changes of the two types of wires during the stretching process (on the VHB substrate) and the bending process (on the PI substrate). In the stretching test, the VHB substrate was stretched to a 1000% stretching rate at a speed of 1 mm/s; in the bending test, the PI substrate was bent to 90° with a bending radius of 5 mm, and one bending cycle takes 5 s. An infrared camera (FOTRIC 220 s, Fotric Smart Technology Co., Ltd.) was used to measure the temperature distribution of the heating circuit on wings. All our replicates are biological replicates, meaning that different samples were prepared for measurement.

### Morphology characterization

A contact angle measuring instrument (SDC-200, Shengding Precision Instrument Co., Ltd) was used to measure the contact angles of liquid metal droplets on various substrates. To characterize the adhesion between semi-liquid metal and substrate, the critical slip angles were tested on a platform with an adjustable incline angle, and recorded using a camera (EOS 200D II, Canon Inc.). High-resolution optical images and element distribution of the patterned semi-liquid metal

circuits were obtained using environmental scanning electron microscope (Apreo, Thermo Fisher Technology Co., Ltd). The complex patterned semi-liquid metal circuits were observed using an optical microscope (FHD Camera V2, Shenzhen Shunhuali Electronics Co., Ltd.). 3D images and cross-sections were obtained using a laser confocal microscope (Olympus, LEXT OLS 4000). The valence states of elements were characterized by XPS (ESCALAB 250Xi, Thermo Fisher Scientific, Oxford, UK).

## Circuits

LEDs (1206) and a microcontroller (MSP430G2553) were pressed onto PDMS substrate to fabricate a running light circuit. LEDs (1206) and touch switch chip (TTP223-BA6) were pressed onto a wooden table in the house model. An analog front-end amplifier (BMD101, NeuroSky Electronic Technology Co., Ltd) was used to collect ECG signals. An analog front-end amplifier (ADS1298, Texas Instruments, Inc., Dallas, TX, USA) was used to collect EMG signals. All procedures involving the attachment of electrodes to human skin comply with ethical guidelines, which have been approved by Tianjin University (approval number: TJUE2025-H-S-063). All subjects voluntarily involved in experiments after informed consent.

## Statistics and reproducibility

All experiments were repeated independently with similar results at least three times.

## Reporting summary

Further information on research design is available in the Nature Portfolio Reporting Summary linked to this article.

## Data availability

The data generated in this study are provided in the Supplementary Information/Source Data file. Source data are provided with this paper.

## Code availability

The code supporting the findings of this study is available from the corresponding authors upon request.

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

## Acknowledgements
This work was supported by the National Natural Science Foundation of China (62304150) (R.G.).

## Author contributions
R.G. and J.L. conceived and designed the project; X.L., T.L., Y.L., C.J., Y.C., Z.Z. and H.Z. performed the experiments; X.L., J.G. and T.L. analyzed the data. R.G., X.L., J.L. wrote and revised the article.

## Competing interests
The authors declare no competing interests.
