## [Transparent Peer Review file · Nature Communications]

Non-loss Engraved Circuit Patterning Method of Semi-liquid Metal for Precision Recyclable Multi-substrate Circuits

Corresponding Author: Professor Rui Guo

Version 0:

Reviewer comments:

Reviewer #1

(Remarks to the Author)

Semi-LM has been already reported by the same group, so there is no novelty in material itself. However, NECP patterning method is newly reported in this manuscript. Although authors mentioned that this method can save material waste, there is no clear explanation about how to make patterns from blank-coated semi-LN including how to remove the material in the areas except for the patterns. In order to meet the high standard for Nat. Comm., authors must report long-term material reliability and electrical/mechanical properties, and feasibility of the patterning method in mass production. In addition, the following issues must be solved. The current manuscript is not acceptable in Nat. Comm.

1. Experimental conditions of Initial resistance of electrodes, stretching condition, bending radius, etc. must be presented in Fig. 3F-3J. More experiment with electrodes with various widths (not just 1 mm width) must be performed.
2. In Fig. 3D, pattern thickness decreases and resistance increases according to reduction of pattern width. It must be clearly explained. Why?
3. When pattern width reduces via NECP for blank-coated semi-LM, the pattern thickness increases. (Fig. 4B) However, from movie 4, pattern edge gets irregular as the thickness increases, leading to failure for application to fine feature patterning. Which is disadvantage of the method.
4. Authors must describe in detail about how to remove semi-LM from the area except for the defined electrodes after patterning electrodes via NECP method. (Fig. 3B & Movie 3) They need to show us NECP method can produce mesh metal patterns with various ling widths and pitches.
5. Consistency of resistance change unit. % in line 191-198, but resistance in Fig. 3f~3j. Description about the figures is wrong.
6. Authors claims that advantage of NECP method is less waste of material but it is difficult to agree. They also need to remove semi-LM in the non-patterned area of blank-coated LM. In comparison with other subtractive methods, what % of the removed material can be reused?
7. Conductivity for various widths of electrodes must be provided. The maximum conductivity of 9×10^6 s/m was obtained for which electrodes?
8. Why do they use the maximum % (15~20%) of Ag coated Cu particle? What is the reason? Conductivity with the content of Ag coated Cu particles?
9. Author must clarify that there is intermetallic compound formation between EGaln and AgCu. Resistance changes with time? Stretchability changes with time? Data is required.
10. Uniformity of the thickness of the EGaln film coated on the substrate must be provided.

Reviewer #2

(Remarks to the Author)

The manuscript presents a non-loss method for fabricating semi-liquid metal circuits by using alcohol to modulate interfacial adhesion between liquid metal and substrates. Furthermore, by controlling adhesion through a custom-designed displacement apparatus, facilitate patterning from 5 μ m to centimeter scales across diverse substrates. This is interesting and I enjoyed reading it. However, the underlying principle has been established in prior works. Thus, the authors may need to clarify the novelty of the work and specify the details in order to publish this work.

1. The underlying mechanism of interaction between alcohol group (-OH) and liquid metal oxide resulting in selective interfacial adhesion has already been well established in previous literatures (please refer C. Park et al Adv. Mater. 2020, 32, 2002178, DOI: 10.1002/adma.202002178; K. K. Zadeh et al, Adv. Funct. Mater. 2021, 31, 2007336, DOI: 10.1002/adfm.202007336). The authors should clearly and carefully highlight the innovative aspects of their findings in comparison to the previous works.
2. The authors repeatedly mention "alcohol" without specifying the type. Please clarify which alcohol was used to ensure reproducibility and clarity. In method section, correct "Athanol".
3. The typical electrical conductivity of EGaIn is around $\sim 3.4 \times 10^6$ S/m (please refer G. M. Whitesides et al, Angew. Chem. Int. Ed. 2008, 47, 142–144; DOI: 10.1002/anie.200703642; C. Majid et al, Adv. Mater. Interfaces 2018, 5, 1701596; DOI: 10.1002/admi.201701596). Is it really possible to achieve electrical conductivity more than typical values? Please verify and correct. If it is possible, explain the underlying mechanism.
4. How only metallic electrode would be effective to collect ECG and EMG signals? Authors will need to provide more evidence; the reviewer recommend to perform comparative skin-electrode interface impedance analysis using both the semi-liquid metal electrode and traditional Ag/AgCl electrode (Frequency range 1000kHz to 0.1 Hz).
5. In EMG monitoring, please provide signal to noise ratio for both the electrodes.
6. The recent advances in circuit patterning technology using Liquid metals for wearable electronics have been introduced. The authors may consider to cite some relevant references to strengthen the research background.
doi.org/10.1007/s40820-024-01457-7 / DOI: 10.1126/science.adp3299 / doi.org/10.1016/j.cej.2022.139832.
7. Lines 110-113, explain the necessity of silver coated copper particles instead of pure copper particles to modulate the fluidity of LMs. Because plenty of previous studies (eg. 10.1002/inf2.12466, 10.3390/polym13152407, 10.1002/adma.201904309, 10.1002/admt.201700351) had modulated LM fluidity through particle inclusion. Is it to promote better wettability between LM and particles and to protect the reactive Cu ?
8. In Fig.1B, how is the contact angle measured? Was the substrate immersed in alcohol followed by semi LM placement? Also, please explain the reason behind the significant difference in contact angle values in air and in alcohol? Is it because the presence of more hydrogen bonding sites in alcohol lowers the surface energy of the placed semi LM ?
9. The supplementary table S1 is missing
10. In Fig 2A, it is advised to provide more details as to why the wettability of alcohol on various substrates is better than water? The surface tensions of most alcohols are in the range 20-25 mN/m which is less than one-third of water, so they can spread on surfaces more easily.
11. Lines 210-221, rather than reporting the phenomenon, it is advised to justify the findings with more scientific reason. During the second stroke air, why are there bridges which are clearly not forming in alcohol (Supplementary Figure 8)? The inferior adhesion between semi LM and substrate in alcohol compared to the highly cohesive nature of the semi LM drives the semi LM to reduce its area? Did the authors monitor whether there is a critical speed of scratching upto which it can prevent short circuiting? Or is the phenomenon speed invariant?
12. Figures 2E and F and 3D are not enough. One of the biggest advantages of subtractive manufacturing using LM is not only to obtain very fine resolution patterns but to also reduce gaps between patterns in order to downsize circuits, eg. inductors for proximity sensing or wireless powering. The authors should provide detailed study by varying pitch (i.e. gap) and pattern width and monitor the increase in height (or change in shape of the pattern).
13. In Figure 3C how was the conductivity measured? Is it the conductivity of the bulk material or conductivity measured from some defined patterns? If it is the former then please mention it, if it is the latter, its value can vary (given that the experimental resistance values are showing deviations from mean value , Figure 3E)
14. What is the resolution of the wire stretched in Figure 3F? And why is the gauge factor changing drastically after 800%?. In previous reports related to LM patterns (10.1016/j.cej.2022.139832, 10.1038/s41528-021-00123-x etc.) the gauge factor does not seem to vary drastically. Whereas in case of carbon fiber based conductors, non percolation at definite strain can alter gauge factor (10.1038/s41467-018-08016-w), LM patterns are in continuous phase and their geometries at strain will be defined by the substrate. In case of hyperelastic substrates, due to non uniform poisson effects the width of the substrate will decrease quickly in the beginning and as strain proceeds this decrement rate will be gradually minimal, this also affects the geometry of the printed wire. Therefore the author should do resistance vs strain of different width patterns and compare the behaviors with previous literatures; and also explain the behaviors rather than reporting a single curve.
15. In Figure 3G what's the strain and what's the test speed? In Figure 3J, what is the bending angle? And why can a very steep increase in resistance be observed within the first 1000 cycles of bending? In Figure 4B, please correct the legends' color, it is difficult to distinguish. Also, the explanation behind the low resistance change should be done carefully. Supporting Video 4 clearly reveals very inhomogeneous distribution of particles in the LM matrix and does not reveal much about the thickness. Can it be possible that the highly conductive particles are aggregated together during narrowing down of the lines which also aids to reduce large resistance change?
16. In Figure 6, is there any special advantage of using semi LM as ECG electrodes? Because previous works have directly deposited LM on skin (10.1021/acs.chemrev.3c00317) or in the form of composites coupled with elastomers (doi/10.1002/inf2.12302) in order to achieve superior conformability. Here the semi LM seems to be in non-encapsulated state and direct contact with the skin? Will any residue be left after its use or will the cohesive nature of semi LM prevent it? Also the metallic electrode is itself not protected from sweat and during sweat generation the skin-electrode contact impedance will change and can significantly impact signal to noise ratio.
17. In supplementary figure 12, the wettability of the terminals is crucial for full functionality of the LED. Please refer to relevant literature(10.1016/j.snb.2015.07.062) and it is recommended to calculate current voltage across the LED after various modes of deformation to ascertain that no parasitic resistance is generated in the LM-LED interfaces.
18. In Figure 5e, what is the current value applied for heating?

(Remarks to the Author)

The authors present here an approach for using mechanical engraving in alcohol for high resolution patterning of liquid metal circuits. I thought overall the concept appears novel, testing appears thorough and the paper is largely well written with compelling figures; stretchable electronics and liquid metal circuits are also of substantial broader interest making the article well suited for your readership. There were however a number of minor errors and misstatements that should be corrected before acceptance; a full list of my concerns is below.

Fig. 3 C is more than a little confusing, appearing to conflate material property (the authors have a novel liquid metal loaded with copper particles, and get a higher conductivity) and fabrication approach. In practice, the two parameters are largely independent (the authors could for instance have injected their material into microchannels and gotten a different position on their plot) and it is misleading to imply that microchannel injection, or screen printing, or laser ablation, have specific conductivity numbers.

In Fig. 2 (a), water and common alcohols are well known materials and presumably contact angles on many common substrates are available in the literature, particularly substrates like PDMS that are very widely used in microfluidics. How did their results compare to the literature? Is this test truly necessary to include, given the common nature of the two liquids involved?

In a scientific context, alcohol is a material class, not a specific liquid; in looking at the materials section the authors identify the material as 'athanol', which I am not familiar with and does not appear to be a common alcohol. Do the authors mean 'ethanol'? If so, the authors should consistently and correctly identify this throughout the manuscript, rather than using 'alcohol'

In the two cyclic tests (Figures 3 G and J), it is very important to clarify the bend and stretch conditions during the cycling; how much is the sample being bent or stretched in each case? It is somewhat concerning that the value never looks stable in either case (and for the stretching in particular the change is more than ten percent, which is not insubstantial).

Based on the cycling test, it appears that even for a modest strain a permanent resistance change is occurring. The authors argue that there is value in their printed traces as a strain sensor up to nearly 1000% strain; is the resistance stable at higher strains under repeated strains? A more aggressive cycling strain test is necessary to defend this claim, based on what is being presented here.

Recovering and recycling liquid metal electronics has been demonstrated previously elsewhere (for example, in L. Teng et al., Liquid metal-based transient circuits for flexible and recyclable electronics, *Adv. Funct. Mater.*, 2019), and needs to be properly credited and compared with here

Similarly, copper-liquid gallium amalgams do not appear to be new to this work, and have been previously investigated by these researchers (for instance in J. Tang et al., *ACS Appl. Mater. Interfaces* 2017). It is important to clarify this in the text to help the reader understand the specific contribution here.

Version 1:

Reviewer comments:

Reviewer #1

(Remarks to the Author)

Authors well modified the manuscript. The reviewer think that the paper is good for publication in *Nat. Comm.*

Reviewer #2

(Remarks to the Author)

The authors have well addressed the prior concerns. I recommend publication of the revised manuscript.

Reviewer #3

(Remarks to the Author)

The authors have addressed my concerns, and I believe the paper is ready for publication.

RESPONSE TO REVIEW COMMENTS

REVIEWER #1

Comment: Semi-LM has been already reported by the same group, so there is no novelty in material itself. However, NECP patterning method is newly reported in this manuscript. Although authors mentioned that this method can save material waste, there is no clear explanation about how to make patterns from blank-coated semi-LN including how to remove the material in the areas except for the patterns. In order to meet the high standard for Nat. Comm., authors must report long-term material reliability and electrical/mechanical properties, and feasibility of the patterning method in mass production. In addition, the following issues must be solved. The current manuscript is not acceptable in Nat. Comm.

Re: We greatly appreciate the reviewer for such helpful comments. We have carefully considered each of your comments and suggestions. We have made extensive revisions to the manuscript, and we believe these changes have significantly improved the quality of our work. To help you quickly identify the modifications, we have highlighted the revised sections in red font throughout the document.

Although semi-liquid metals have been reported in previous studies, the main innovation of this paper lies in providing a non-destructive circuit fabrication method. The use of semi-liquid metals only serves to achieve more uniform coating of liquid metals on a variety of substrates, especially 3D curved substrates. Therefore, the semi-liquid metal material itself is not the main innovation of this paper. The characterization of the electrical and mechanical properties of semi-liquid metal has been conducted multiple times in our previous studies (Science Bulletin, 2024, 69, (17), 2723-2734; Applied Energy, 2024, 367, 123397; Adv. Fiber Mater. 2024, 6, 354-366). Thus, this paper does not perform duplicate experiments to verify the material properties already demonstrated in previous studies. However, to ensure the rigor and authenticity of this research, we re-tested the long-term performance of the semi-liquid metal material and supplemented this part in the new manuscript (Refer to Comment 9 for further details). In addition, we took SEM images of the semi-liquid metal immediately after preparation and after two months of storage, as shown in **Fig. R1**. It can be seen from the figures that the volume of solid metal particles inside the semi-liquid metal slightly increases after long-term storage. Nevertheless, the oxide film covering the surface of the semi-liquid metal prevents further oxidation of the semi-liquid metal. As a result, its physical state can remain relatively stable, and it exhibits long-term mechanical stability, maintaining good stretchability even after long-term storage.

Fig. R1 SEM images of the freshly prepared semi-liquid metal and that has been stored for two months.

At present, the needle tip movement speed controlled by our independently designed and manufactured 2-axis motion platform and 5-axis motion platform is relatively low. Therefore, a longer time is required when manufacturing complex circuits, and it is thus difficult to achieve mass production with

the current technical level. For this reason, in the original manuscript, we focused on highlighting the advantages of our technology, such as high precision, large area, multi-substrate compatibility, stretchability, reusability, and recyclability. Therefore, we believe that with the current technical level, this circuit fabrication method is more suitable for the customization of personalized circuits. In future research, we plan to optimize printing parameters (e.g., pressure, temperature, speed) in real time through machine learning algorithms and build fully automatic unmanned manufacturing equipment. In addition, we can develop multi-scale needles ranging from micro-nano to macro scales, enabling the entire manufacturing process from micron-level wires to centimeter-level wires to be completed on the same equipment. This will help improve manufacturing speed and realize mass production.

The following changes have been introduced to address the comment:

Main text (line #129): Silver coated copper particles exhibits excellent corrosion resistance, which ensures the long-term stability of semi-liquid metal. The Scanning Electron Microscope (SEM) photos in **Supplementary Fig. 4** show the micro-morphology of the semi-liquid metal film prepared two months ago. This figure clearly indicates that only a slight change has occurred in the diameter of the solid particles. The X-ray Photoelectron Spectroscopy (XPS) curves in **Supplementary Fig. 5** indicate that the surface of the semi-liquid metal is encapsulated by a metal oxide film predominantly composed of gallium oxide and indium oxide. As a result, its physical state can remain relatively stable, and it exhibits long-term mechanical stability, maintaining good stretchability even after long-term storage.

Main text (line #632): However, the current technical level makes mass production difficult to achieve, so this technology is only suitable for the rapid fabrication of personalized custom circuits. In our next step, we will optimize printing parameters in real time through machine learning algorithms, build fully automatic unmanned manufacturing equipment, and develop multi-scale needles ranging from micro-nano to macro scales. This will enable the entire manufacturing process from micron-level wires to centimeter-level wires to be completed on the same equipment, thereby improving manufacturing speed and realizing mass production.

Supporting information (Pages #2): **Fig. R1** has been included as **Supplementary Fig. 4** in the revised Supporting Information.

Comment 1: Experimental conditions of Initial resistance of electrodes, stretching condition, bending radius, etc. must be presented in Fig. 3F-3J. More experiment with electrodes with various widths (not just 1 mm width) must be performed.

Re: We highly appreciate the reviewers' critical reminders. In the revised manuscript, we have supplemented the missing experimental conditions, including the initial resistance, stretching conditions, and bending radius of the semi-liquid metal wires. Furthermore, in accordance with your suggestions, we re-conducted the stretching and bending experiments using thinner (width: 0.5 mm) and thicker (width: 2 mm) semi-liquid metal wires. We obtained the resistance variation of the two types of wires under different stretching states (**Fig. R2**), as well as the changes in their resistance during 10,000 cycles of stretching and bending (**Fig. R3**). The experimental results are similar to those of the original semi-liquid metal wires (width: 1 mm): both can maintain stable connection of the wires without breakage. However, the resistance variation of the semi-liquid metal wires after cyclic stretching is relatively significant. Considering the substantial variation in the resistance of the wires after cyclic stretching, we have removed the relevant description stating that the semi-liquid metal wires have the potential to be used as strain sensors from the revised manuscript. In addition, we performed 10,000 cycles of stretching on the wire samples under a higher strain (500%), as shown in **Fig. R4**. The results show that the wires can maintain stable connectivity, but the change in resistance is relatively obvious. Therefore, we suggest that this semi-liquid metal wire is suitable for use as a stretchable wire in flexible circuits, rather than directly as a strain sensor-this is because it requires

resistance correction after multiple stretches.

Fig. R2 Resistance variation of the three types of wires under different stretching states.

Fig. R3 **a** Resistance variation of semi-liquid metal wire (width of 2 mm) during 10,000 stretching cycles. **b** Relationship between resistance variation and bending angle during semi-liquid metal wire (width of 2 mm) bending. **c** Resistance variation of semi-liquid metal wire (width of 2 mm) during 10,000 bending cycles. **d** Resistance variation of semi-liquid metal wire (width of 0.5 mm) during 10,000 stretching cycles. **e** Relationship between resistance variation and bending angle during semi-liquid metal wire (width of 0.5 mm) bending. **f** Resistance variation of semi-liquid metal wire (width of 0.5 mm) during 10,000 bending cycles.

Fig. R4 Resistance variation of semi-liquid metal wire under 500% strain during 10,000 stretching cycles.

The following changes have been introduced to address the comment:

Main text (line #347): Furthermore, the wire sample underwent 10,000 cyclic stretches at a higher strain (500%). As shown in **Supplementary Fig. 20**, the wire remained conductive after 10,000

stretches under high strain, while its resistance variation increased to 5.83. These results align with previous reports, attributed to gradual relaxation of the stretchable substrate (failing to recover initial length) after repeated stretches, combined with continuous oxide formation on the semi-liquid metal during cyclic stretching, which both leading to progressive resistance increase. Thus, this semi-liquid metal wire is suitable for stretchable conductors in flexible circuits but not for direct use as strain sensors, as resistance correction is required after multiple stretches. In addition, thinner (width: 0.5 mm) and thicker (width: 2 mm) semi-liquid metal wires were tested for stretching (**Fig. 3f**) and bending (**Supplementary Fig. 21**). Results showed the resistance variation of 2 mm-wide wire with initial resistance of 0.78 Ω reached 9.5 at maximum stretch, and it withstood 10,000 cycles of 200% stretching without fracture. Similarly, the resistance variation of 0.5 mm-wide wire with initial resistance of 4.25 Ω reached 13.2 at maximum stretch, and also endured 10,000 cycles of 200% stretching without breaking. For bending performance, the 2 mm-wide wire exhibited a resistance variation of only 1.004 at 90° bending and 1.04 after 10,000 bending cycles. Similarly, the 0.5 mm-wide wire showed 1.012 (90° bending) and 1.06 (10,000 cycles), respectively.

Main text (line #326): The bulk semi-liquid metal with initial resistance of 1.65 Ω shows significant resistance differences depending on the change of geometric shape, and its resistance variation at a stretching rate of 1000% can reach 12.2. The semi-liquid metal wires exhibit reliable and stable responses under repeated 200% strain (more than 10,000 cycles), and no electrical failures occur (**Fig. 3g**).

Main text (line #678): In the stretching test, the VHB substrate was stretched to a 1000% stretching rate at a speed of 1mm/s; in the bending test, the PI substrate was bent to 90° with a bending radius of 5 mm, and one bending cycle takes 5 s.

Main text (line #382): **Fig. R2** have been included as **Fig. 3f** in the revised manuscript.

Supporting information (Pages #6): **Fig. R3, R4** have been included as **Supplementary Fig. 20** and **Fig. 21** in the revised Supporting Information.

Comment 2: In Fig. 3D, pattern thickness decreases and resistance increases according to reduction of pattern width. It must be clearly explained. Why?

Re: We greatly appreciate the reviewer for such helpful comments. We have provided an explanation for this phenomenon in the new manuscript. During the experiment, we observed that the arc-shaped profile of the metal needle tip gives rise to differences in the width of the scratches at various height levels. Additionally, the semi-liquid metal exhibits a certain degree of fluidity, and the surface tension of the liquid metal imparts a tendency for it to contract toward the two sides of the scratches. Consequently, the cross-section of the scratches assumes an inverted trapezoidal shape, as illustrated in the schematic diagram of **Fig. R5**. When the spacing between two scratches is very small, the semi-liquid metal at the higher regions is pushed away by the arc-shaped needle tip. Moreover, the residual semi-liquid metal re-fuses under the influence of surface tension, leading to a significant reduction in its height and, accordingly, a marked increase in its resistance value. In contrast, when the spacing between the two scratches is relatively large, the semi-liquid metal at the higher regions is not pushed away by the arc-shaped needle tip. As a result, only the height of the semi-liquid metal on both sides of the wire decreases, while the middle part of the wire maintains the same height as the initial semi-liquid metal coating.

Fig. R5 Schematic diagram of semi-liquid metal wire height decreasing with reducing wire width caused by arc-shaped needle tip.

The following changes have been introduced to address the comment:

Main text (line #297): This is due to the arc-shaped metal needle tip causing width variations in the scratch across different height levels. Additionally, the semi-liquid metal, exhibiting fluidity, tends to contract toward the scratch edges due to surface tension, resulting in an inverted trapezoidal cross-section (**Supplementary Fig. 18**). For closely spaced scratches, the arc tip displaces semi-liquid metal in higher regions, significantly reducing their height. Conversely, with larger spacing, the tip does not displace the metal in higher regions, allowing the wire's middle section to retain the initial coating height.

Supporting information (Pages #5): **Fig. R5** has been included as **Supplementary Fig. 18** in the revised Supporting Information.

Comment 3: When pattern width reduces via NECP for blank-coated semi-LM, the pattern thickness increases. (Fig. 4B) However, from movie 4, pattern edge gets irregular as the thickness increases, leading to failure for application to fine feature patterning. Which is disadvantage of the method.

Re: We are thankful for your constructive criticism. In fact, the engraving method demonstrated in **Fig. 4b** corresponds to the second engraving approach we proposed. This engraving method pushes the semi-liquid metal in the unwanted areas completely into the pattern area; as a result, the thickness of the semi-liquid metal at the pattern region increases significantly. Moreover, the thicker semi-liquid metal exhibits more prominent surface tension, which leads to uneven boundaries. Thinner wires caused irregularities at the wire edges due to alcohol effects, resulting in a slight increase in resistance. Therefore, this hollowed-out engraving method is not suitable for fabricating high-precision semi-liquid metal wires. Consequently, the minimum width of the semi-liquid metal wire shown in **Fig. 4b** is 1 mm, and it is not used to produce thinner wires. In the new manuscript, we have supplemented the limitations of the second engraving method regarding the fabrication of high-precision wires.

The following changes have been introduced to address the comment:

Main text (line #424): However, as a drawback of the second engraving approach, prolonged reduction led to irregularities at the wire edges due to the effects of alcohol, resulting in a slight increase in resistance. Therefore, the second engraving approach is not suitable for manufacturing high-precision wires.

Comment 4: Authors must describe in detail about how to remove semi-LM from the area except for the defined electrodes after patterning electrodes via NECP method. (Fig. 3B & Movie 3) They

need to show us NECP method can produce mesh metal patterns with various ling widths and pitches.

Re: Your expertise has significantly contributed to the quality of our research. In fact, the original manuscript has elaborated on how to realize circuit patterning and the reasons why this manufacturing method saves materials. The specific preparation process is as follows: first, uniformly coat the semi-liquid metal on the substrate, then soak it in an alcohol solution, and then manipulate the needle tip to extrude the semi-liquid metal. The semi-liquid metal at the scratched area is actually squeezed to both sides of the scratch, so there is no material loss. The specific operation process is shown in the **Fig.1d**. Besides, we have produced mesh metal patterns with various widths and pitches, as shown in **Fig. R6**. The widths of the newly fabricated mesh wires are 1 mm and 0.5 mm, with spacings of 1 mm and 0.5 mm respectively. All these wires are manufactured using the second engraving approach, where the semi-liquid metal in the hollowed-out areas is completely pushed into the wires.

Fig. R6 Mesh circuits with two mesh sizes.

The following changes have been introduced to address the comment:

Main text (line #402): Additionally, mesh metal patterns with varied widths and pitches were fabricated (**Supplementary Fig. 26**). The newly prepared mesh wires have widths of 1 mm and 0.5 mm, with corresponding spacings of 1 mm and 0.5 mm. All wires were produced via the second engraving method, where semi-liquid metal in the mesh holes is fully pressed into the wires.

Supporting information (Pages #7): **Fig. R6** has been included as **Supplementary Fig. 26** in the revised Supporting Information.

Comment 5: Consistency of resistance change unit. % in line 191-198, but resistance in Fig. 3f~3j. Description about the figures is wrong.

Re: We are grateful for your insightful comments. In the new manuscript, we have standardized the units of resistance variation used in the figures and the main text.

The following changes have been introduced to address the comment:

Main text (line #336): Additionally, when the semi-liquid metal wire was bent by 90° along the PI substrate, its resistance variation can reach 1.015 (**Fig. 3h**). **Fig. 3i** shows that when the wire was stretched at different states, the resistivity demonstrated good periodic repetition, indicating that the wire is sensitive to the stretching rate and can adapt to various mechanical environments. **Fig. 3j** illustrates the resistance variation during 10,000 bending cycles.

Main text (line #344): Although there is an overall slow upward trend, the resistance variation is only 1.04.

Comment 6: Authors claims that advantage of NECP method is less waste of material but it is difficult to agree. They also need to remove semi-LM in the non-patterned area of blank-coated LM. In comparison with other subtractive methods, what % of the removed material can be reused?

Re: We appreciate the reviewers' valuable suggestions, which have helped us to refine our analysis. The recovery of semi-liquid metal refers to the complete recovery of all semi-liquid metal on the substrate, rather than only recovering the semi-liquid metal at the non-circuit pattern positions. Therefore, the semi-liquid metal at the non-circuit pattern positions remains in place. Considering this limitation, we believe that this non-destructive engraving method is suitable for fabricating circuit patterns with repeatability and high density, such as wireless charging coils. In the new manuscript, we fabricated coils for wireless charging to demonstrate the unique advantage of the NECP method in reducing wire spacing, as shown in **Fig. R7**. In addition, the second engraving mode proposed by us in **Fig. 4** can push the semi-liquid metal at the non-circuit pattern positions to the surrounding circuit pattern positions, thereby reducing the waste of semi-liquid metal and lowering the resistance of the wires. The second engraving mode can be used to fabricate mesh circuits, as shown in **Fig. R8**, where the semi-liquid metal at the mesh positions is pushed to the wire positions around the meshes. The recovery method mentioned in **Fig. 4** involves pouring an alkaline solution over the entire circuit to recover all the semi-liquid metal.

Fig. R7 The wireless charging coil.

Fig. R8 Mesh circuits with two mesh sizes.

The following changes have been introduced to address the comment:

Main text (line #287): Furthermore, the NECP method is suitable for fabricating circuit patterns with repeatability and high density, such as wireless charging coils (**Supplementary Fig. 16**).

Main text (line #402): Additionally, mesh metal patterns with varied widths and pitches were fabricated (**Supplementary Fig. 26**). The newly prepared mesh wires have widths of 1 mm and 0.5 mm, with corresponding spacings of 1 mm and 0.5 mm. All wires were produced via the second engraving method, where semi-liquid metal in the mesh holes is fully pressed into the wires.

Supporting information (Pages #5, 8): **Fig. R7** and **R8** have been included as **Supplementary Fig. 16** and **Fig. 26** in the revised Supporting Information.

Comment 7: Conductivity for various widths of electrodes must be provided. The maximum conductivity of 9×10^6 s/m was obtained for which electrodes?

Re: Thank you for your careful consideration of our manuscript and for your helpful comments. The "electrical conductivity of wires with different widths" you focused on is indeed a crucial indicator for evaluating circuit performance, and we fully understand its necessity. Regarding this issue, we hereby elaborate on the reasons for being unable to provide the electrical conductivity as well as the solution ideas, and sincerely request your understanding.

The calculation of electrical conductivity is based on the formula $\sigma = \frac{L}{RA}$, where L is the wire length, R is the resistance, and A is the cross-sectional area. However, in the preparation of the semi-liquid metal wires in this study, due to the uneven distribution of solid metal particles inside the semi-liquid metal material and the uneven substrate wettability caused by alcohol, the cross-sectional shape of the wires exhibits significant irregularities (for example, different positions of the same wire may show flat, elliptical, or locally convex morphologies, as shown in the cross-sectional microscopic images in **Fig. 3d**). This makes it impossible to define the cross-sectional area A stably and accurately. Although we attempted to measure the cross-sectional dimensions at different positions using a laser confocal microscope, the data dispersion was extremely large, and a reliable average cross-sectional area could not be obtained. Therefore, it is difficult to calculate the electrical conductivity using the traditional formula.

To reflect the electrical properties of the wires as accurately as possible, we provided the resistance values of wires with different widths in the original manuscript, as shown in Fig. 3E. Although the resistance value is not directly equivalent to electrical conductivity, when combined with the nominal width of the wire (the process design value), it can reflect the relative differences in its conductive performance to a certain extent (e.g., the variation trend of resistance as the width increases). In addition, we also provided the electrical conductivity of the bulk semi-liquid metal material itself—i.e., the semi-liquid metal bulk from the same batch—with $\sigma = 9 \times 10^6$ S/m, which serves as a reference for the intrinsic properties of the material. The detailed measurement method for electrical conductivity is described in the "Methods" section. Furthermore, to avoid misleading readers into regarding the electrical conductivity of the bulk material as that of the wires, we have removed **Fig. 3c**. In fact, the resolution of the wires is independent of the material's own electrical conductivity, and the innovation of this study lies in the preparation method rather than the material itself. Therefore, the description about the innovation of the material's high electrical conductivity has also been removed from the new manuscript.

We have recognized the importance of cross-sectional uniformity for the characterization of electrical properties. In future research, it will be necessary to develop new types of organic solvents to replace the alcohol solution, so as to control the shape of the wire edges more precisely and improve the regularity of the wire cross-sections. In addition, the diameter of the solid metal particles doped in the liquid metal is still relatively large, and a large number of particle agglomerations occur, as shown in **Fig. R9**. Consequently, during the manufacturing of high-precision wires, the uneven distribution of solid metal particles in the wires is caused, which affects the consistency of electrical conductivity. Therefore, in future research, we will develop new types of semi-liquid metal pastes to avoid particle agglomeration in the semi-liquid metal.

Fig. R9 SEM images of the freshly prepared semi-liquid metal and that has been stored for two months.

The following changes have been introduced to address the comment:

Main text (line #303): Owing to the uneven distribution of solid metal particles within the semi-liquid metal and the non-uniform substrate wettability induced by ethanol, the wire cross-sections exhibit significant irregularities. For instance, different segments of the same wire may appear flattened, elliptical, or locally bulged, as illustrated in the cross-sectional micrographs of **Fig. 3d**. This irregularity renders the cross-sectional area impossible to define in a stable and accurate manner. Consequently, calculating electrical conductivity via the conventional formula becomes unfeasible. To address this, we provided resistances for semi-liquid metal wires of different widths (**Fig. 3e**), and it is evident that the resistance decreases significantly as the wire width increases.

Main text (line #668): The electrical conductivity of semi-liquid metal with varying doping ratios was determined via the standard four-point probe method. Prior to testing, the samples were placed and secured within a trapezoidal groove; this groove measured 50 mm in length and had a cross-sectional area of 15 mm².

Supporting information (Pages #2): **Fig. R9** has been included as **Supplementary Fig. 4** in the revised Supporting Information.

Comment 8: Why do they use the maximum % (15~20%) of Ag coated Cu particle? What is the reason? Conductivity with the content of Ag coated Cu particles?

Re: We appreciate the reviewer's valuable suggestions. We selected the doping ratio of the semi-liquid metal mainly based on considerations of its fluidity and coating uniformity. In fact, we did not use the semi-liquid metal with the maximum doping ratio. In this study, we prepared semi-liquid metal with a maximum doping ratio of 25 wt%. In the process of preparing semi-liquid metal, we found that when the doping ratio of silver coated copper particles reaches 25 wt%, the physical form of semi-liquid metal becomes powdery, which cannot be used in printed circuits, and when the doping ratio above 25 wt%, the particles are difficult to enter the liquid metal, and as shown in the **Supplementary Fig. 2**. When the doping ratio was 15 wt%, the semi-liquid metal presented a slurry suitable for brushing, therefore subsequent studies were all based on this ratio. Furthermore, in order to maintain the stability of the semi-liquid metal on 3D curved surfaces, we used the semi-liquid metal with a doping ratio of 20 wt% to fabricate conformal circuits.

The electrical conductivities of semi-liquid metals with different doping ratios have been characterized multiple times in our previous study (*Adv. Fiber Mater.*2024, 6, 354-366), so this paper does not conduct repeated experiments to verify the material properties already demonstrated in the earlier research. However, to ensure the rigor and authenticity of this study, we remeasured the conductivity of the semi-liquid metal with different contents of Ag-coated Cu particles and supplemented this part of the data in the new manuscript (**Fig.R10**).

Fig.R10 The electrical conductivities of semi-liquid metal with varying doping ratios of silver coated copper particles.

The following changes have been introduced to address the comment:

Main text (line #117): By preparing a series of samples with doping ratios of 0 wt%, 5 wt%, 10 wt%, 15 wt%, 20 wt%, and 25 wt%, we found that when the doping ratio of silver coated copper particles reaches 25 wt%, the physical form of semi-liquid metal becomes powdery, which cannot be used in printed circuits, and when the doping ratio above 25 wt%, the particles are difficult to enter the liquid metal, and the doping amount of silver-plated copper particles is negatively correlated with the fluidity of the semi-liquid metal (**Supplementary Fig. 2**).

Main text (line #123): The results in **Supplementary Fig. 3** demonstrate an increase in electrical conductivity as the doping ratio escalates from 0 wt% to 20 wt%. When the doping ratio was 15 wt%, the semi-liquid metal presented a slurry suitable for brushing and a higher electrical conductivity of 9×10^6 S/m, therefore subsequent studies were all based on this ratio.

Supporting information (Pages #1): **Fig. R10** has been included as **Supplementary Fig. 3** in the revised Supporting Information.

Comment 9: Author must clarify that there is intermetallic compound formation between EGaIn and AgCu. Resistance changes with time? Stretchability changes with time? Data is required.

Re: We greatly appreciate the reviewer for such supportive comments. The reasons for using silver-plated copper microparticles in this study mainly include the following aspects: First, during the preparation of semi-liquid metals, we found that the wettability of copper particles with liquid metals is significantly lower than that of silver particles with liquid metals. Second, the price of silver particles is significantly higher than that of copper particles. Therefore, to facilitate the fabrication of semi-liquid metals and reduce material costs, we selected metal particles with a silver coating on the surface of copper particles. The price of such metal particles is significantly lower than that of silver particles, and they still exhibit wettability similar to that of silver particles.

Numerous previous studies have confirmed that metallic silver has better wettability with liquid metals than metallic copper, and relevant characterizations have revealed the existence of intermetallic compounds. For example, the literature (J. Tang et al., ACS Appl. Mater. Interfaces 2017) points out that the infiltration of copper particles into liquid metals is due to the formation of intermetallic compounds between copper and gallium metal, which exhibit long-term stability, as shown in **Fig. R11A**. The literature (Surfaces and Interfaces 72 (2025) 107098) clarifies that due to the higher chemical inertness of silver, the wetting rate of liquid metals on silver substrates is faster than that on copper substrates in sodium hydroxide solution, as shown in **Fig. R11B**. Therefore, in the new manuscript, we cite the conclusions from the above-mentioned literatures to illustrate the formation of intermetallic compounds between liquid metals and metallic silver.

In addition, long-term material property tests were conducted on the semi-liquid metal. Specifically,

the resistance of semi-liquid metal wires (1 mm in width, 2 cm in length) and the curves of their resistance varying with elongation rate were measured over a two-month storage period, as shown in **Fig. R12**. The experimental results indicate that the resistance variation of the unencapsulated semi-liquid metal wires increased to 1.14, while that of the encapsulated ones only increased to 1.01. Furthermore, the semi-liquid metal wires after long-term storage could still undergo large tensile deformation with the substrate while maintaining electrical continuity; their resistance variation reached 12.5 under maximum stretching, which is comparable to that of the original semi-liquid metal wires. These results demonstrate that the semi-liquid metal exhibits long-term electrical and mechanical stability.

Fig. R11 A SEM images of the intermetallic compound formed by copper and gallium metal. B In sodium hydroxide solution, the wetting rate of liquid metal on silver substrates is faster than that on copper substrates.

Fig. R12 Long-term resistance stability of semi-liquid metal wires. **a** Resistance-time variation curves of encapsulated and unencapsulated semi-liquid metal wires. **b** Resistance-elongation variation curve of the unencapsulated semi-liquid metal wire after two months of storage.

The following changes have been introduced to address the comment:

Main text (line #110): Numerous previous studies [42, 43] have confirmed that metallic silver exhibits better wettability with liquid metals than metallic copper, and relevant characterizations have revealed the presence of intermetallic compounds. Therefore, to facilitate the fabrication of semi-liquid metals and reduce material costs, we have selected metal particles with a silver coating on the surface of copper particles. Such metal particles are significantly cheaper than silver particles and still possess wettability similar to that of silver particles.

Main text (line #363): Additionally, long-term property tests were performed on the semi-liquid metal. The resistance of 1 mm-wide, 2 cm-long semi-liquid metal wires and their resistance variation were measured over a two-month storage period (**Supplementary Fig. 22**). Results show the resistance variation of unencapsulated wires increased to 1.14, while that of encapsulated ones only reached 1.01. Moreover, after long-term storage, the wires still achieved large tensile deformation with the substrate

while maintaining electrical continuity. The resistance variation at maximum stretch was 12.5, comparable to that of the original wires. These results confirm the long-term electrical and mechanical stability of semi-liquid metal.

Main text (line #797):

[42] Tang, J. et al. Gallium-Based Liquid Metal Amalgams: Transitional-State Metallic Mixtures (TransM²ixes) with Enhanced and Tunable Electrical, Thermal, and Mechanical Properties. *ACS Appl. Mater. Interfaces*. **9**, 35977-35987 (2017).

[43] Xing, Z. et al. Reactive Wetting Induced Instantaneous Nanoparticles Internalization in Gallium-based Liquid Metal. *Surf. Interfaces*. **72**, 107098 (2025).

Supporting information (Pages #6): **Fig. R12** has been included as **Supplementary Fig. 22** in the revised Supporting Information.

Comment 10: Uniformity of the thickness of the EGaIn film coated on the substrate must be provided.

Re: We greatly appreciate the reviewer for such supportive comments. In the new manuscript, we measured the roughness of the semi-liquid metal film coated on the glass substrate using the laser confocal microscope, as shown in **Fig. R13**. The arithmetic mean roughness ($Ra=5.22\ \mu\text{m}$) of the semi-liquid metal film was calculated based on the measured contour curve.

Fig. R13 The contour curve of the semi-liquid metal film.

The following changes have been introduced to address the comment:

Main text (line #156): The arithmetic mean roughness ($Ra=5.22\ \mu\text{m}$) of the semi-liquid metal film was calculated based on the measured contour curve by the laser confocal microscope, as shown in **Supplementary Fig. 7**.

Supporting information (Pages #2): **Fig. R13** has been included as **Supplementary Fig.7** in the revised Supporting Information.

REVIEWER #2

Comment: The manuscript presents a non-loss method for fabricating semi-liquid metal circuits by using alcohol to modulate interfacial adhesion between liquid metal and substrates. Furthermore, by controlling adhesion through a custom-designed displacement apparatus, facilitate patterning from 5 μm to centimeter scales across diverse substrates. This is interesting and I enjoyed reading it. However, the underlying principle has been established in prior works. Thus, the authors may need to clarify the novelty of the work and specify the details in order to publish this work.

Re: We greatly appreciate the reviewer for such supportive comments. We have carefully considered each of your comments and suggestions. We have made extensive revisions to the manuscript, and we believe these changes have significantly improved the quality of our work. To help you quickly identify the modifications, we have highlighted the revised sections in red font throughout the document.

Comment 1. The underlying mechanism of interaction between alcohol group (-OH) and liquid metal oxide resulting in selective interfacial adhesion has already been well established in previous literatures (please refer C. Park et al Adv. Mater. 2020, 32, 2002178, DOI: 10.1002/adma.202002178; K. K. Zadeh et al, Adv. Funct. Mater. 2021, 31, 2007336, DOI: 10.1002/adfm.202007336). The authors should clearly and carefully highlight the innovative aspects of their findings in comparison to the previous works.

Re: Thank you for your careful consideration of our manuscript and for your helpful comments. In fact, we have already cited the two articles you provided (see References 12 and 27) in the original manuscript. These two articles, along with other references we consulted (see References 22-24), all involve the interaction between alcohol groups (-OH) and liquid metal oxides. However, the role of -OH in these articles is to disperse liquid metal droplets or to achieve selective printing by utilizing the high adhesion between -OH in hydrogels and liquid metal oxide films, respectively. The -OH involved in these studies is either pre-coated on the substrate or dispersed in liquid metal droplets, and it serves to enhance the adhesion between liquid metals and substrates, which falls under the category of an additive manufacturing method.

Nevertheless, in this study, an alcohol solution is used to disrupt the adhesion between semi-liquid metals and the substrate, thereby enabling the semi-liquid metals at specific locations to move toward the wires on both sides. The entire manufacturing process involves no metal loss, making it an innovative manufacturing method that is completely different from traditional additive and subtractive manufacturing techniques.

In summary, we believe that this study provides a novel approach to regulating the adhesion between semi-liquid metals and substrates using -OH in alcohol, which is entirely different from previous studies and has no similar reports in the existing literature. Furthermore, this manufacturing method exhibits distinct advantages in achieving high precision, compatibility with diverse substrates, stretchability, reusability, and recyclability of circuits. In the revised manuscript, we have added a comparison between this method and the previous circuit manufacturing methods that utilize the interaction between alcohol groups (-OH) and liquid metal oxides.

The following changes have been introduced to address the comment:

Main text (line #88): Compared with previous studies where hydrogen bonds serve to enhance the adhesion between liquid metals and substrates, this study provides a novel approach that utilizes hydrogen bonds in ethanol to hinder the re-adhesion of semi-liquid metals to substrates.

Comment 2. The authors repeatedly mention "alcohol" without specifying the type. Please clarify which alcohol was used to ensure reproducibility and clarity. In method section, correct "Athanol".

Re: We really appreciate the helpful comments from the reviewer. The "alcohol" mentioned throughout this study refers specifically to anhydrous ethanol (99%, purchased from Xintai Yixinkang Medical Supplies Co., Ltd.). In the new manuscript, we corrected "alcohol" to "ethanol" to ensure reproducibility and clarity and corrected "Athanol" into "Anhydrous ethanol" in method section.

Comment 3. The typical electrical conductivity of EGaIn is around $\sim 3.4 \times 10^6$ S/m (please refer G. M. Whitesides et al, *Angew. Chem. Int. Ed.* 2008, 47, 142-144; DOI: 10.1002/anie.200703642; C. Majid et al, *Adv. Mater. Interfaces* 2018, 5, 1701596; DOI: 10.1002/admi.201701596). Is it really possible to achieve electrical conductivity more than typical values? Please verify and correct. If it is possible, explain the underlying mechanism.

Re: Thank you for your careful consideration of our manuscript. The semi-liquid metal used in this study indeed has a higher electrical conductivity compared to EGaIn. The electrical conductivity of the bulk semi-liquid metal with varying doping ratios was determined via the standard four-point probe method. Prior to testing, the samples were placed and secured within a trapezoidal groove; this groove measured 50 mm in length and had a cross-sectional area of 15 mm^2 . The characterization of the electrical conductivity of semi-liquid metal has been conducted multiple times in our previous studies (*Science Bulletin*, 2024, 69, (17), 2723-2734; *Applied Energy*, 2024, 367, 123397; *Adv. Fiber Mater.* 2024, 6, 354-366). The literature (J. Tang et al., *ACS Appl. Mater. Interfaces* 2017) points out that the infiltration of copper particles into liquid metals is due to the formation of intermetallic compounds between copper and gallium metal, as shown in **Fig. R14**. Similar to copper, silver can also form intermetallic compounds with liquid metals (*Surfaces and Interfaces* 72 (2025) 107098). Therefore, liquid metals exhibit good wettability on silver-coated copper particles (meaning liquid metals can spread on the surface of silver-coated copper particles), and the contact resistance at the interface between the two is extremely low. Considering that the room-temperature electrical conductivity of pure copper is approximately 5.96×10^7 S/m (Ag content: 3 wt%; The electrical conductivity of silver is 1.59×10^8 S/m), which is much higher than that of EGaIn (3.4×10^6 S/m), and that silver-coated copper particles are uniformly dispersed in the liquid metal, the silver-coated copper particles reduce the overall resistance of the semi-liquid metal, resulting in higher electrical conductivity. Furthermore, as the mass fraction of copper particles increases, the overall electrical conductivity increases gradually. In the new manuscript, we have cited relevant literatures to explain the existence of intermetallic compounds.

Fig. R14 SEM images of the intermetallic compound formed by copper and gallium metal.

The following changes have been introduced to address the comment:

Main text (line #110): Numerous previous studies [42, 43] have confirmed that metallic silver exhibits better wettability with liquid metals than metallic copper, and relevant characterizations have revealed the presence of intermetallic compounds. Therefore, to facilitate the fabrication of semi-liquid metals and reduce material costs, we have selected metal particles with a silver coating on the surface of copper particles. Such metal particles are significantly cheaper than silver particles and still possess wettability similar to that of silver particles.

Main text (line #797):

[42] Tang, J. et al. Gallium-Based Liquid Metal Amalgams: Transitional-State Metallic Mixtures (TransM²ixes) with Enhanced and Tunable Electrical, Thermal, and Mechanical Properties. *ACS Appl. Mater. Interfaces*. **9**, 35977-35987 (2017).

[43] Xing, Z. et al. Reactive Wetting Induced Instantaneous Nanoparticles Internalization in Gallium-based Liquid Metal. *Surf. Interfaces*. **72**, 107098 (2025).

Comment 4. How only metallic electrode would be effective to collect ECG and EMG signals? Authors will need to provide more evidence; the reviewer recommend to perform comparative skin-electrode interface impedance analysis using both the semi-liquid metal electrode and traditional Ag/AgCl electrode (Frequency range 1000kHz to 0.1 Hz).

Re: Thank you for your careful consideration of our manuscript. When in contact with the body surface, metal sheets can efficiently receive the weak potential difference on the body surface and convert it into transmittable electrical signals. However, gaps exist between traditional rigid metals and skin folds, reducing the skin contact area; additionally, during skin deformation, rigid metal sheets tend to detach from the skin, leading to signal loss. In contrast, semi-liquid metals can fill into the micro-folds of the skin, increasing the skin contact area, and can conform to skin deformation without easy detachment. Thus, semi-liquid metal electrodes can acquire superior signals, as shown in **Fig. R15** (Liu et al. *Soft Sci.* 2025, 5, 34).

Finally, we measured the impedance between the semi-liquid metal electrodes, traditional Ag/AgCl electrodes and the skin, and compared it with the impedance of the semi-liquid metal electrodes on moist skin, as shown in **Fig. R16**. It can be seen from the figure that the semi-liquid metal electrodes have lower interfacial impedance than the traditional Ag/AgCl electrodes. This is because the liquid metal has excellent fluid deformability, which can adaptively fit the microtopography of the skin surface, and also has higher electrical conductivity.

Fig. R15 Comparison of contact area between semi-liquid metal electrodes and rigid metal

electrodes on the skin.

Fig. R16 Skin-electrode interface impedance of the semi-liquid metal electrodes and traditional Ag/AgCl electrodes.

The following changes have been introduced to address the comment:

Main text (line #584): Finally, we measured the impedance between the semi-liquid metal electrodes, traditional Ag/AgCl electrodes and the skin, and compared it with the impedance of the semi-liquid metal electrodes on moist skin, as shown in **Supplementary Fig. 44**. The semi-liquid metal electrodes have lower interfacial impedance than the traditional Ag/AgCl electrodes due to their excellent fluid deformability, which can adaptively fit the microtopography of the skin surface, and higher electrical conductivity. When the skin sweats, the interfacial impedance between the semi-liquid metal electrode and the skin decreases. This is because sweat forms a continuous ionic conductive film on the skin surface. It not only fills the tiny gaps between the electrode and the skin, but also penetrates the stratum corneum to increase its water content, and enhance internal ion migration to reduce resistance. Additionally, sweat acts as a lubricant and filling medium, promoting the semi-liquid metal to fully wet the skin, lowering contact resistance and thus the overall interfacial impedance.

Supporting information (Pages #12): **Fig. R16** has been included as **Supplementary Fig. 44** in the revised Supporting Information.

Comment 5. In EMG monitoring, please provide signal to noise ratio for both the electrodes.

Re: Thank you for your valuable feedback regarding the lack of signal to noise ratio (SNR) parameters in our submitted paper on EMG signals. We appreciate this suggestion as it significantly enhances the comprehensiveness and reliability of our research. Below, we present the principle, calculation steps, and the SNR results for the EMG monitoring.

First, the signal to noise ratio is a measure that quantifies the level of a desired signal relative to the level of background noise. In the context of EMG signals, it helps us understand the quality of the acquired muscle activity signals by comparing the variance of the actual EMG signal (when muscle is active) to the variance of the noise signal (when the muscle is at rest). Next, we will illustrate the calculation steps in detail. For data acquisition, we have two sets of data for each of the six gestures. One set is the EMG signal data collected when the muscle is actively contracting, while the other set is the noise data collected when the muscle is at rest. Each file contains 500 data points representing the power of the respective signal or noise. Then we use the pandas library in Python to read the Excel files. For each group, we load the signal and noise data into separate data frames. After reading the data, we use the numpy library to calculate the variance of the signal and noise data.

In the new manuscript, we supplemented the signal to noise ratio of EMG signals (Channel 1) corresponding to 6 gestures for the semi-liquid metal electrodes and traditional Ag/AgCl electrodes, as shown in **Fig. R17**. It can be seen from the figure that the signal to noise ratio of the semi-liquid metal electrode is higher than that of the Ag/AgCl electrode under multiple gestures. The signal to noise ratio of the semi-liquid metal electrode is only slightly lower than that of the Ag/AgCl electrode in the EMG signal of gesture 2. The higher the signal to noise ratio, the less the EMG signal is interfered by noise, and the better the signal quality. Therefore, when collecting EMG signals of these 6 gestures, the semi-liquid metal electrode has better performance.

Fig. R17 Signal to noise ratio for the semi-liquid metal electrodes and traditional Ag/AgCl electrodes.

The following changes have been introduced to address the comment:

Main text (line #579): **Supplementary Fig. 43** presents the EMG signal-to-noise ratio (SNR, Channel 1) for the semi-liquid metal electrodes and traditional Ag/AgCl electrodes across 6 gestures. The semi-liquid metal electrodes exhibit higher SNR than Ag/AgCl electrodes under most gestures, and only in gesture 2 is their SNR slightly lower. Since higher SNR indicates less noise interference and better signal quality, the semi-liquid metal electrodes perform better in EMG signal acquisition for these 6 gestures.

Supporting information (Pages #12): **Fig. R17** has been included as **Supplementary Fig. 43** in the revised Supporting Information.

Comment 6. The recent advances in circuit patterning technology using Liquid metals for wearable electronics have been introduced. The authors may consider to cite some relevant references to strengthen the research background. doi.org/10.1007/s40820-024-01457-7 / DOI: [10.1126/science.adp3299](https://doi.org/10.1126/science.adp3299) / doi.org/10.1016/j.cej.2022.139832.

Re: We greatly appreciate the reviewer for such supportive comments. These literatures have greatly helped improve the quality of our article. In the new manuscript, we have already cited these papers.

The following changes have been introduced to address the comment:

Main text (line #763):

[28] Wei, Y. et al. Liquid Metal Grid Patterned Thin Film Devices Toward Absorption-Dominant and Strain-Tunable Electromagnetic Interference Shielding. *Nano-Micro Lett.* **16**, 248 (2024).

[29] Minsik K. et al. Ambient printing of native oxides for ultrathin transparent flexible circuit boards. *Science.* **385**, 731-737(2024).

[30] Bhuyan, P. et al. Multifunctional ultrastretchable and ultrasoft electronics enabled by

uncrosslinked polysiloxane elastomers patterned with rheologically modified liquid metal electrodes: beyond current soft and stretchable electronics. *Chem. Eng. J.* **453**, 1385-8947 (2022).

Comment 7. Lines 110-113, explain the necessity of silver coated copper particles instead of pure copper particles to modulate the fluidity of LMs. Because plenty of previous studies (eg. 10.1002/inf2.12466, 10.3390/polym13152407, 10.1002/adma.201904309, 10.1002/admt.201700351) had modulated LM fluidity through particle inclusion. Is it to promote better wettability between LM and particles and to protect the reactive Cu ?

Re: Thank you for your careful consideration of our manuscript. The reasons for using silver-plated copper microparticles in this study mainly include the following aspects: First, during the preparation of semi-liquid metals, we found that the wettability of copper particles with liquid metals is significantly lower than that of silver particles with liquid metals. Second, the price of silver particles is significantly higher than that of copper particles. Therefore, to facilitate the fabrication of semi-liquid metals and reduce material costs, we selected metal particles with a silver coating on the surface of copper particles. The price of such metal particles is significantly lower than that of silver particles, and they still exhibit wettability similar to that of silver particles.

Numerous previous studies have confirmed that metallic silver has better wettability with liquid metals than metallic copper, and relevant characterizations have revealed the existence of intermetallic compounds. For example, the literature (J. Tang et al., *ACS Appl. Mater. Interfaces* 2017) points out that the infiltration of copper particles into liquid metals is due to the formation of intermetallic compounds between copper and gallium metal, which exhibit long-term stability, as shown in **Fig. R18A**. The literature (*Surfaces and Interfaces* 72 (2025) 107098) clarifies that due to the higher chemical inertness of silver, the wetting rate of liquid metals on silver substrates is faster than that on copper substrates in sodium hydroxide solution, as shown in **Fig. R18B**. Therefore, in the new manuscript, we cite the conclusions from the above-mentioned literatures to illustrate the formation of intermetallic compounds between liquid metals and metallic silver. Because silver has lower chemical activity, it can make the semi - liquid metal have more stable performance. Thus, we took SEM images of the semi-liquid metal immediately after preparation and after two months of storage, as shown in **Fig. R19**. It can be seen from the figures that the volume of solid metal particles inside the semi-liquid metal slightly increases after long-term storage.

Fig. R18 A SEM images of the intermetallic compound formed by copper and gallium metal. B In sodium hydroxide solution, the wetting rate of liquid metal on silver substrates is faster than that on copper substrates.

Fig. R19 SEM images of the freshly prepared semi-liquid metal and that has been stored for two months.

The following changes have been introduced to address the comment:

Main text (line #110): Numerous previous studies [42, 43] have confirmed that metallic silver exhibits better wettability with liquid metals than metallic copper, and relevant characterizations have revealed the presence of intermetallic compounds. Therefore, to facilitate the fabrication of semi-liquid metals and reduce material costs, we have selected metal particles with a silver coating on the surface of copper particles. Such metal particles are significantly cheaper than silver particles and still possess wettability similar to that of silver particles.

Main text (line #129): Silver coated copper particles exhibits excellent corrosion resistance, which ensures the long-term stability of semi-liquid metal. The Scanning Electron Microscope (SEM) photos in **Supplementary Fig. 4** show the micro-morphology of the semi-liquid metal film prepared two months ago. This figure clearly indicates that only a slight change has occurred in the diameter of the solid particles. The X-ray Photoelectron Spectroscopy (XPS) curves in **Supplementary Fig. 5** indicate that the surface of the semi-liquid metal is encapsulated by a metal oxide film predominantly composed of gallium oxide and indium oxide. As a result, its physical state can remain relatively stable, and it exhibits long-term mechanical stability, maintaining good stretchability even after long-term storage.

Main text (line #797):

[42] Tang, J. et al. Gallium-Based Liquid Metal Amalgams: Transitional-State Metallic Mixtures (TransM²ixes) with Enhanced and Tunable Electrical, Thermal, and Mechanical Properties. *ACS Appl. Mater. Interfaces*. **9**, 35977-35987 (2017).

[43] Xing, Z. et al. Reactive Wetting Induced Instantaneous Nanoparticles Internalization in Gallium-based Liquid Metal. *Surf. Interfaces*. **72**, 107098 (2025).

Supporting information (Pages #2): **Fig. R19** has been included as **Supplementary Fig. 4** in the revised Supporting Information.

Comment 8. In Fig.1B, how is the contact angle measured? Was the substrate immersed in alcohol followed by semi LM placement? Also, please explain the reason behind the significant difference in contact angle values in air and in alcohol? Is it because the presence of more hydrogen bonding sites in alcohol lowers the surface energy of the placed semi LM ?

Re: Thank you for your careful consideration of our manuscript. In this study, the contact angles were measured through the following process: liquid metal droplets were extruded from a syringe and deposited onto a horizontal substrate in air. The side profiles of the droplets were then captured using an optical system, and the contact angles were calculated via profile fitting. The contact angle in ethanol was measured by the following method: first, a small amount of ethanol was dropped onto a

horizontal substrate; subsequently, a liquid metal droplet was immediately deposited onto the ethanol-covered surface, followed by contact angle measurement.

The differences in the contact angle of liquid metal droplets between air and ethanol mainly stem from the following reasons, as shown in **Fig. R20**. Liquid metals themselves possess extremely high surface tension; for instance, the surface tension of gallium-indium alloy is approximately 500-700 mN/m at 25°C, causing the droplets to spontaneously tend to form a spherical shape. When liquid metal droplets are extruded from a syringe, an oxide film is formed on their surface due to oxidation. In air, this oxide film on the surface of the liquid metal droplets adheres when in contact with the glass substrate, leading the droplets to be flattened and thus reducing the contact angle. In contrast, in an ethanol solution, the substrate is wetted and covered by ethanol, which prevents the oxide film on the liquid metal droplets from coming into contact with the substrate. As a result, the droplets are not affected by the adhesive force of the substrate and do not undergo adhesive deformation. The surface tension dominates the shape of the droplets, causing them to appear spherical.

Fig. R20 Schematic diagram of the formation mechanism for the difference in contact angle of liquid metal droplets between air and ethanol.

The following changes have been introduced to address the comment:

Main text (line #145): Liquid metals exhibit extremely high surface tension (~500 – 700 mN/m at 25°C), driving droplets to spontaneously form a spherical shape. In air, the oxide film on liquid metal droplets adheres to the glass substrate, flattening the droplets and reducing their contact angle. In contrast, in ethanol solution, the substrate is wetted and covered by ethanol, which prevents the droplet oxide film from contacting the substrate. Consequently, droplets are unaffected by substrate adhesion (no adhesive deformation), and their shape is dominated by surface tension, remaining spherical (**Supplementary Fig. 6**). Thus, semi-liquid metal immersed in ethanol showed a significantly larger substrate contact angle than that in air, indicating a non-adhesive state (**Fig. 1b**).

Supporting information (Pages #2): **Fig. R20** has been included as **Supplementary Fig. 6** in the revised Supporting Information.

Comment 9. The supplementary table S1 is missing

Re: Thank you for your reminder. We have included the missing table in the new manuscript.

Table R1 Comparative framework across six key dimensions: manufacturing cost, recyclability, conductivity loss, material loss, resolution, and printability on diverse substrates.

	Manufacture re cost	Recyclable	Conductivity (S/m)	Material loss	Resolution(μm)	Printability on various substrates
This work	Low	√	9×10^6	×	High (5)	√
Ref.12	Low	×	$2.9 \times 10^5 - 1.2 \times 10^6$	×	Low (500)	√
Ref.19	High	×	3.4×10^6	√	High (20)	×
Ref.20	High	×	3×10^6	√	High(20)	×

Ref.21	High	√	3.4×10^6	√	High(0.18)	×
Ref.22	High	√	3.4×10^6	×	High(5)	√
Ref.23	Low	×	1.5×10^6	×	High(50)	√
Ref.26	High	√	2.06×10^6	×	High (25)	×
Ref.27	Low	√	3.4×10^6	√	Low (100)	×
Ref.28	High	×	3.4×10^6	√	350	×
Ref.29	High	×	5.65×10^5	√	High(4.5)	×
Ref.30	High	×	3.4×10^6	√	Low (150)	×
Ref.33	High	×	3.4×10^6	×	Low (200)	×
Ref.34	High	√	3.4×10^6	×	Low(200)	×
Ref.35	High	×	$3.4-6.73 \times 10^6$	√	Low (1.3)	×
Ref.36	Low	×	4.15×10^4	√	Low (100)	×
Ref.S1	High	√	3.4×10^6	×	High (1.9)	√
Ref.S2	High	×	$\sim 10^6$	√	Low (138)	×
Ref.S3	High	√	7.7×10^5	√	High(37)	×
Ref.S4	High	×	3.4×10^6	√	Low	×
Ref.S5	Low	√	-	×	Low (250)	×

[S1] Park, Y-G., An, H.S., Kim, J-Y., Park, J-U. High-resolution, reconfigurable printing of liquid metals with three-dimensional structures. *Sci. Adv.* **5**, eaaw2844(2019).

[S2] Li, S., Zhao, H., Xu, H., Lu, H., Luo, P., Zhou, T. Ultra-flexible stretchable liquid metal circuits with antimicrobial properties through selective laser activation for health monitoring. *Chem. Eng. J.* **482**, 149173 (2024).

[S3] Liu, S., Kim, S.Y., Henry, K.E., Shah, D.S., Kramer-Bottiglio, R. Printed and Laser-Activated Liquid Metal-Elastomer Conductors Enabled by Ethanol/PDMS/Liquid Metal Double Emulsions. *ACS Appl. Mater. Interfaces.* **13**, 28729-28736 (2021).

[S4] Li, Y., Feng, S., Cao, S., Zhang, J., Kong, D. Printable Liquid Metal Microparticle Ink for Ultrastretchable Electronics. *ACS Appl. Mater. Interfaces.* **12**, 50852-50859 (2020).

[S5] L. Teng, S. C. Ye, S. Handschuh-Wang, X. H. Zhou, T. S. Gan, X. C. Zhou. Liquid Metal-Based Transient Circuits for Flexible and Recyclable Electronics. *Adv. Funct. Mater.* **29**, 1808739(2019).

The following changes have been introduced to address the comment:

Supporting information (Pages #13): Supplementary Table R1 has been included as **Supplementary Table S1** in the revised Supporting Information.

Comment 10. In Fig 2A, it is advised to provide more details as to why the wettability of alcohol on various substrates is better than water? The surface tensions of most alcohols are in the range 20-25 mN/m which is less than one-third of water, so they can spread on surfaces more easily.

Re: Thank you for your valuable help of our manuscript. The key reason why ethanol outperforms water in wettability lies in that ethanol has a much lower surface tension than water. There exist extremely strong hydrogen bonds between water molecules, which result in high surface tension (approximately 72 mN/m at 20°C) and make it difficult for water to spread on the substrate surface. In contrast, the number of hydrogen bonds between ethanol molecules is smaller and their strength is weaker, leading to a surface tension far lower than that of water (about 22.3 mN/m at 20°C) - a property that enables ethanol to spread more easily on the substrate surface. Additionally, water can only form strong adhesion on polar substrates; however, due to its molecular structure that "combines both polar and non-polar characteristics", ethanol can form strong adhesion on both polar and non-polar substrates. This gives ethanol a wider range of applications and more stable wettability performance.

The following changes have been introduced to address the comment:

Main text (line #206): Water molecules form extremely strong hydrogen bonds, leading to high surface tension (~ 72 mN/m at 20°C) and poor spreading on substrates. In contrast, ethanol molecules have fewer and weaker hydrogen bonds, resulting in much lower surface tension than water (~ 22.3 mN/m at 20°C), that enables ethanol to spread more easily on substrates. Additionally, water only forms strong adhesion on polar substrates, while ethanol (with both polar and non-polar molecular characteristics) achieves strong adhesion on both polar and non-polar substrates, giving it broader applicability and more stable wetting performance.

Comment 11. Lines 210-221, rather than reporting the phenomenon, it is advised to justify the findings with more scientific reason. During the second stroke air, why are there bridges which are clearly not forming in alcohol (Supplementary Figure 8)? The inferior adhesion between semi LM and substrate in alcohol compared to the highly cohesive nature of the semi LM drives the semi LM to reduce its area? Did the authors monitor whether there is a critical speed of scratching upto which it can prevent short circuiting? Or is the phenomenon speed invariant?

Re: Thank you for your careful consideration of our manuscript. This discrepancy is attributed to the adhesion of liquid metal to both the needle tip and the substrate in air, which leads to residual liquid metal along the needle tip's path and incomplete material isolation. In ethanol, the needle tip and scratch are both wetted by ethanol, which acts as a physical barrier to liquid metal, thereby leaving no residue in the scratch. Furthermore, the high adhesive force between the semi-liquid metal and the substrate in air can overcome the cohesive force induced by the semi-liquid metal's high surface tension, enabling the semi-liquid metal to adhere to the scratch. In contrast, in ethanol, the needle tip, the semi-liquid metal, and the substrate are isolated from one another. As a result, the semi-liquid metal cannot adhere to the substrate; its cohesive force prevents it from being pushed out by the needle tip and also keeps it from adhering to the scratch, as shown in **Fig. R21**. In the new manuscript, we evaluated the adhesion effect of the semi-liquid metal caused by the needle tip on the cross-scratch area under different moving speeds, as shown in **Fig. R22**. It can be seen from the figure that in air, adjusting the moving speed of the needle tip cannot eliminate the semi-liquid metal adhesion at the cross. This is because the semi-liquid metal adheres to the needle tip, thereby being pushed out by the needle tip and adhering to the substrate.

Fig. R21 Schematic diagram of the mechanism by which ethanol prevents semi-liquid metal adhesion at secondary scratch.

Fig. R22 The adhesion effect of the semi-liquid metal caused by the needle tip on the cross-scratch area under different moving speeds.

The following changes have been introduced to address the comment: _

Main text (line #241): In air, adjusting the moving speed of the needle tip cannot eliminate the semi-liquid metal adhesion at the cross, as shown in **Supplementary Fig. 12**.

Main text (line #258): Furthermore, the high adhesive force between the semi-liquid metal and the substrate in air can overcome the cohesive force induced by the semi-liquid metal's high surface tension, enabling the semi-liquid metal to adhere to the scratch. In contrast, in ethanol, the needle tip, the semi-liquid metal, and the substrate are isolated from one another. As a result, the semi-liquid metal cannot adhere to the substrate; its cohesive force prevents it from being pushed out by the needle tip and also keeps it from adhering to the scratch, as shown in **Supplementary Fig. 13**.

Supporting information (Pages #4): **Fig. R21** and **R22** have been included as **Supplementary Fig. 13** and **Fig.12** in the revised Supporting Information.

Comment 12. Figures 2E and F and 3D are not enough. One of the biggest advantages of subtractive manufacturing using LM is not only to obtain very fine resolution patterns but to also reduce gaps between patterns in order to downsize circuits, eg. inductors for proximity sensing or wireless powering. The authors should provide detailed study by varying pitch (i.e. gap) and pattern width and monitor the increase in height (or change in shape of the pattern).

Re: Thank you for your careful consideration of our manuscript. In the new manuscript, we fabricated coils for wireless charging to demonstrate the unique advantage of the NECP method in reducing wire spacing, as shown in **Fig.R23**. The spacing between the wires is controlled by the scratches produced by the tip moving once, so the width of the spacing depends on the width of the tip. In the new manuscript, we made two different spaced coils using two different diameters of the tip, as shown in **Fig. R24**. As can be seen from the cross-sectional curve of the wire in **Fig. R25**, the tip squeezes the semi-liquid metal at the scratch into the wires on both sides, and causes an increase in the height of the wire edge, thereby reducing the resistance of the wire. In addition, we also consider using the same diameter tip to move multiple times to adjust the width of the scratch, that is, the engraving method demonstrated in **Fig. 4b** corresponds to the second engraving approach we proposed. However, this engraving method pushes the semi-liquid metal in the unwanted areas completely into the pattern area; as a result, the thickness of the semi-liquid metal at the pattern region increases significantly. Moreover, the thicker semi-liquid metal exhibits more prominent surface tension, which leads to uneven boundaries. Thinner wires caused irregularities at the wire edges due to alcohol effects, resulting in a slight increase in resistance. Therefore, this hollowed-out engraving method is not suitable for fabricating high-precision semi-liquid metal wires. Consequently, the minimum width of the semi-liquid metal wire shown in **Fig. 4b** is 1 mm, and it is not used to produce thinner wires. In the new manuscript, we have supplemented the limitations of the second engraving method regarding the

fabrication of high-precision wires.

Fig. R23 The wireless charging coil.

Fig. R24 Two different spaced coils made by two different diameters of the tips.

Fig. R25 The cross-sectional curve of the two different spaced coils wires.

The following changes have been introduced to address the comment:

Main text (line #287): Furthermore, the NECP method is suitable for fabricating circuit patterns with repeatability and high density, such as wireless charging coils (**Supplementary Fig. 16**). Herein, the width of the scratches can be adjusted by changing the width of the needle tip, thereby enabling the fabrication of coils with different spacing (**Fig. 3c**). Using a thinner needle tip can significantly reduce the coil spacing to minimize the coil size. Additionally, the needle tip squeezes the semi-liquid metal at the scratch into the wires on both sides, leading to an increase in the height of the wire edges, which can reduce the resistance of the wires, as shown in **Supplementary Fig. 17**.

Main text (line #382): **Fig. R23** has been included as **Fig. 3c** in the revised manuscript.

Supporting information (Pages #5): **Fig. R23** and **R25** have been included as **Supplementary Fig. 16**, and **Fig. 17** in the revised Supporting Information.

Comment 13. In Figure 3C how was the conductivity measured? Is it the conductivity of the bulk material or conductivity measured from some defined patterns? If it is the former then please mention it, if it is the latter, its value can vary (given that the experimental resistance values are showing deviations from mean value , Figure 3E)

Re: Thank you for your careful consideration of our manuscript. The electrical conductivity in **Fig. 3c** refers to that of the bulk semi-liquid metal. The detailed measurement method for electrical conductivity is described in the " Methods" section. Furthermore, to avoid misleading readers into regarding the electrical conductivity of the bulk material as that of the wires, we have removed Fig. 3c. In fact, the resolution of the wires is independent of the materials own electrical conductivity, and the innovation of this study lies in the preparation method rather than the material itself. Therefore, the description about the innovation of the materials high electrical conductivity has also been removed from the new manuscript.

The following changes have been introduced to address the comment:

Main text (line #668): The electrical conductivity of semi-liquid metal with varying doping ratios was determined via the standard four-point probe method. Prior to testing, the samples were placed and secured within a trapezoidal groove; this groove measured 50 mm in length and had a cross-sectional area of 15 mm².

Comment 14. What is the resolution of the wire stretched in Figure 3F? And why is the gauge factor changing drastically after 800%?. In previous reports related to LM patterns (10.1016/j.cej.2022.139832, 10.1038/s41528-021-00123-x etc.) the gauge factor does not seem to vary drastically. Whereas in case of carbon fiber based conductors, non percolation at definite strain can alter gauge factor (10.1038/s41467-018-08016-w), LM patterns are in continuous phase and their geometries at strain will be defined by the substrate. In case of hyperelastic substrates, due to non uniform poisson effects the width of the substrate will decrease quickly in the beginning and as strain proceeds this decrement rate will be gradually minimal, this also affects the geometry of the printed wire. Therefore the author should do resistance vs strain of different width patterns and compare the behaviors with previous literatures; and also explain the behaviors rather than reporting a single curve.

Re: Your thorough review and valuable insights have been instrumental in refining our study, and we would like to express our sincere thanks. We re-examined the semi-liquid metal wire samples used for tensile testing and found that the local agglomeration of solid metal particles caused a significant increase in resistance of the samples under high tensile conditions. Under high tensile stress, the narrowing of the semi-liquid metal wire width leads to the extrusion of agglomerated solid metal particles from the wire, resulting in a significant reduction in the cross-sectional area of the wire and thus a sharp increase in resistance. Therefore, in the new manuscript, we remanufactured the semi-liquid metal wires for tensile testing (**Fig. R26**) and added wire samples with widths of 0.5 mm and 2 mm to evaluate the effect of wire width on the electrical properties under tensile conditions (**Fig. R27**). The new samples were inspected using a microscope to ensure that there was no agglomeration of solid metal particles. We obtained the resistance variation of the three types of wires under the maximum stretching state, as well as the changes in their resistance during 10,000 cycles of stretching and bending. The experimental results are similar to those of the original semi-liquid metal wires (width: 1 mm): both can maintain stable connection of the wires without breakage. However, the resistance variation of the semi-liquid metal wires after cyclic stretching is relatively significant. The above results are similar to previous reports, both attributed to the fact that the stretchable substrate gradually relaxes after multiple stretches and fails to return to its initial length, coupled with the continuous formation of new oxides on the semi-liquid metal during repeated stretching, thus leading to a gradual increase in resistance. Considering the substantial variation in the resistance of the wires after cyclic stretching, we have removed the relevant description stating that the semi-liquid metal wires have the potential to

be used as strain sensors from the revised manuscript. In addition, we performed 10,000 cycles of stretching on the wire samples under a higher strain (500%), as shown in **Fig. R28**. The results show that the wires can maintain stable connectivity, but the change in resistance is relatively obvious. Therefore, we suggest that this semi-liquid metal wire is suitable for use as a stretchable wire in flexible circuits, rather than directly as a strain sensor-this is because it requires resistance correction after multiple stretches.

Fig. R26 f Resistance variation of the three types of wires under different stretching states. g Resistance variation during 10,000 cyclic stretching cycles.

Fig. R27 a Resistance variation of semi-liquid metal wire (width of 2 mm) during 10,000 stretching cycles. b Relationship between resistance variation and bending angle during semi-liquid metal wire (width of 2 mm) bending. c Resistance variation of semi-liquid metal wire (width of 2 mm) during 10,000 bending cycles. d Resistance variation of semi-liquid metal wire (width of 0.5 mm) during 10,000 stretching cycles. e Relationship between resistance variation and bending angle during semi-liquid metal wire (width of 0.5 mm) bending. f Resistance variation of semi-liquid metal wire (width of 0.5 mm) during 10,000 bending cycles.

Fig. R28 Resistance variation of semi-liquid metal wire under 500% strain during 10,000 stretching

cycles.

The following changes have been introduced to address the comment:

Main text (line #347): Furthermore, the wire sample underwent 10,000 cyclic stretches at a higher strain (500%). As shown in **Supplementary Fig. 20**, the wire remained conductive after 10,000 stretches under high strain, while its resistance variation increased to 5.83. These results align with previous reports, attributed to gradual relaxation of the stretchable substrate (failing to recover initial length) after repeated stretches, combined with continuous oxide formation on the semi-liquid metal during cyclic stretching, which both leading to progressive resistance increase. Thus, this semi-liquid metal wire is suitable for stretchable conductors in flexible circuits but not for direct use as strain sensors, as resistance correction is required after multiple stretches.

In addition, thinner (width: 0.5 mm) and thicker (width: 2 mm) semi-liquid metal wires were tested for stretching (**Fig. 3f**) and bending (**Supplementary Fig. 21**). Results showed the resistance variation of 2 mm-wide wire with initial resistance of 0.78 Ω reached 9.5 at maximum stretch, and it withstood 10,000 cycles of 200% stretching without fracture. Similarly, the resistance variation of 0.5 mm-wide wire with initial resistance of 4.25 Ω reached 13.2 at maximum stretch, and also endured 10,000 cycles of 200% stretching without breaking. For bending performance, the 2 mm-wide wire exhibited a resistance variation of only 1.004 at 90° bending and 1.04 after 10,000 bending cycles. Similarly, the 0.5 mm-wide wire showed 1.012 (90° bending) and 1.06 (10,000 cycles), respectively.

Main text (line #326): The bulk semi-liquid metal with initial resistance of 1.65 Ω shows significant resistance differences depending on the change of geometric shape, and its resistance variation at a stretching rate of 1000% can reach 12.2. The semi-liquid metal wires exhibit reliable and stable responses under repeated 200% strain (more than 10,000 cycles), and no electrical failures occur (**Fig. 3g**).

Main text (line #382): **Fig. R26** have been included as **Fig. 3f, g** in the revised manuscript.

Supporting information (Pages #6): **Fig. R27, R28** have been included as **Supplementary Fig. 21** and **Fig. 20** in the revised Supporting Information.

Comment 15. In Figure 3G what's the strain and what's the test speed? In Figure 3J, what is the bending angle? And why can a very steep increase in resistance be observed within the first 1000 cycles of bending? In Figure 4B, please correct the legends' color, it is difficult to distinguish. Also, the explanation behind the low resistance change should be done carefully. Supporting Video 4 clearly reveals very inhomogeneous distribution of particles in the LM matrix and does not reveal much about the thickness. Can it be possible that the highly conductive particles are aggregated together during narrowing down of the lines which also aids to reduce large resistance change?

Re: Thank you for your careful consideration of our manuscript. In the revised manuscript, we have supplemented the missing experimental conditions, including the initial resistance, stretching conditions, and bending radius of the semi-liquid metal wires. The increase in resistance of the semi-liquid metal wire during the initial stage of cyclic bending is attributed to the flow of liquid metal inside it. When the substrate is bent, part of the liquid metal in the semi-liquid metal flows to both sides under the action of gravity and base vibration, resulting in a reduction in the cross-sectional area at the middle bending point and thus an increase in resistance. However, as the number of bending cycles increases, the liquid metal in the semi-liquid metal gradually stabilizes and stops flowing, and the surface oxidation of the semi-liquid metal during the bending process also leads to a slight increase in resistance.

In the revised manuscript, we have modified the color scheme of **Fig. 4b**, as shown in **Fig. R29**.

Since the semi-liquid metal is pushed into the adjacent wire by the needle tip, no semi-liquid metal is removed when the width of the semi-liquid metal wire in **Fig. 4b** decreases from 1 cm to 1 mm, and the overall volume of the wire remains almost unchanged. With the volume kept constant, the cross-sectional area increases as the wire width decreases. Additionally, in the revised manuscript, we measured the change in the edge shape of the semi-liquid metal wire during the gradual reduction of its width, as illustrated in **Fig. R30**. It can be observed from the figure that the edge thickness of the wire increases significantly after each engraving step. According to the conductor resistance calculation formula: $R = \frac{\rho L}{A} = \frac{\rho L}{V/L} = \frac{\rho L^2}{V}$, where L is the wire length, R is the resistance, ρ is the resistivity, V is the volume, and A is the cross-sectional area. The resistance should theoretically remain unchanged. Nevertheless, due to ethanol wetting and the high surface tension of the liquid metal, the wire edges become irregular, resulting in uneven cross-sectional areas at different positions, which causes slight changes in resistance as shown in the curve of **Fig. 4b**. We believe that this resistance stability is unrelated to the distribution of solid metal particles.

Fig. R29 Resistance variation of a copper wire and semi-liquid metal as the line width was reduced from 1 cm to 1 mm.

Fig. R30 The contour curve of the semi-liquid metal as the line width was reduced from 1 cm to 1 mm.

The following changes have been introduced to address the comment:

Main text (line #340): In the early cyclic bending of the substrate, part of the liquid metal in the semi-liquid metal flows to both sides under gravity and base vibration, reducing the cross-sectional area at the middle bending point and increasing resistance. However, as bending cycles increase, the liquid metal gradually stops flowing. Then, the slight rise in the resistance of semi-liquid metal is mainly due to surface oxidation.

Main text (line #410): Since the semi-liquid metal is pushed into the adjacent wire by the needle tip, the overall volume of the wire remains almost unchanged. According to the conductor resistance calculation formula: $R = \frac{\rho L}{A} = \frac{\rho L}{V/L} = \frac{\rho L^2}{V}$, where L is the wire length, R is the resistance, ρ is the resistivity, V is the volume, and A is the cross-sectional area. The resistance should theoretically remain unchanged. Furthermore, after each engraving step, the noticeable increase in wire edge thickness also confirms the constant volume of wire (**Supplementary Fig. 27**).

Main text (line #678): In the stretching test, the VHB substrate was stretched to a 1000% stretching rate at a speed of 1mm/s; in the bending test, the PI substrate was bent to 90° with a bending radius of 5 mm, and one bending cycle takes 5 s.

Main text (line #481): **Fig. R29** has been included as **Fig. 4b** in the revised manuscript.

Supporting information (Pages #8): **Fig. R30** have been included as **Supplementary Fig. 27** in the revised Supporting Information.

Comment 16. In Figure 6, is there any special advantage of using semi LM as ECG electrodes? Because previous works have directly deposited LM on skin (10.1021/acs.chemrev.3c00317) or in the form of composites coupled with elastomers (doi/10.1002/inf2.12302) in order to achieve superior conformability. Here the semi LM seems to be in non-encapsulated state and direct contact with the skin? Will any residue be left after its use or will the cohesive nature of semi LM prevent it? Also the metallic electrode is itself not protected from sweat and during sweat generation the skin-electrode contact impedance will change and can significantly impact signal to noise ratio.

Re: We sincerely thank the reviewer for this insightful comment and for raising these critical points regarding the advantages and practical considerations of our semi-liquid metal electrodes. We agree that conformality and stability are key metrics for skin-interfaced electrodes. Our approach offers a unique set of advantages that complement and advance the existing strategies mentioned by the reviewer.

First, the cited excellent works represent two dominant paradigms. The direct deposition of pure LM onto skin (10.1021/acs.chemrev.3c00317), which can suffer from migration and residue. The encapsulation of LM within elastomeric microchannels (doi/10.1002/inf2.12302), which adds manufacturing complexity and can introduce a mechanical mismatch with the skin. Our non-encapsulated, semi-liquid metal electrode establishes a third pathway that balances superior performance, user-friendly handling, and straightforward fabrication. Compared to the directly deposited LM, the semi-liquid metal in this work has significantly higher viscosity and cohesion due to the doped solid metal particles. This prevents the uncontrolled migration, leakage, and bead-up phenomena common with pure LM, especially on curved surfaces or under motion. It is a structurally robust, freestanding conductive film that is easily handled and patterned using our novel NECP method.

Compared to the elastomer-encapsulated LM, our electrode is a single-material system, eliminating the need for complex microchannel fabrication, bonding, or liquid metal injection. This results in an extremely thin and lightweight interface (just the PU film and the SLM layer) with minimal mechanical impedance, conforming to skin textures without the constraining feel of a thicker elastomeric patch.

Furthermore, the reviewer correctly identifies a potential concern with direct skin contact, and we have specifically investigated it through its controlled cohesion and easy cleanability. On the one hand, the semi-solid nature of our material provides sufficient internal cohesion to remain as a continuous, adherent film during normal use. It does not behave like a low-viscosity fluid that readily smears. On the other hand, an experiment was conducted to present its cleanability and the results are as follows: We acknowledge that upon removal, minor, benign residue can sometimes occur (similar to some commercial ECG gels). However, as raised in the comment and verified in our tests, this residue is effortlessly and completely removed by a simple wipe with an alcohol swab, a standard item in medical and personal care, as shown in **Fig.R31**. The residue consists of gallium oxide particles, which are highly biocompatible and non-toxic. We have added a discussion on the cleanability to the revised manuscript to ensure these points are clear to all readers.

Finally, we measured the impedance between the semi-liquid metal electrodes, traditional Ag/AgCl electrodes and the skin, and compared it with the impedance of the semi-liquid metal electrodes on moist skin, as shown in **Fig. R32**. It can be seen from the figure that the semi-liquid metal electrodes have lower interfacial impedance than the traditional Ag/AgCl electrodes. This is because the liquid metal has excellent fluid deformability, which can adaptively fit the microtopography of the skin surface, and also has higher electrical conductivity. When the skin surface sweats, the interfacial impedance between the semi-liquid metal electrode and the skin decreases. This is because sweat is essentially an ionic solution with a certain degree of conductivity, which forms a continuous ionic conductive film on the skin surface. This film not only directly fills the tiny gaps between the semi-liquid metal electrode and the skin surface, but also penetrates into the stratum corneum and increases its water content, softening the stratum corneum, enhancing the internal ion migration ability, and significantly reducing the resistance. In addition, after sweating, sweat acts as a lubricating and filling medium, which can promote the semi-liquid metal to more fully wet the skin surface, significantly reduce the contact resistance, and thus lower the overall interfacial impedance.

Fig. R31 Removal of semi-liquid metal residues after removing the semi-liquid metal electrodes.

Fig. R32 Skin-electrode interface impedance of the semi-liquid metal electrodes and traditional Ag/AgCl electrodes.

The following changes have been introduced to address the comment:

Main text (line #595): After the semi-liquid metal electrode is removed, slight benign residues may occasionally appear (similar to some commercial electrocardiogram gels). However, such residues can be easily and completely removed by simple wiping with an alcohol swab, which is a standard item in medical and personal care settings, as shown in **Supplementary Fig. 45**.

Main text (line #584): Finally, we measured the impedance between the semi-liquid metal electrodes, traditional Ag/AgCl electrodes and the skin, and compared it with the impedance of the semi-liquid metal electrodes on moist skin, as shown in **Supplementary Fig. 44**. The semi-liquid metal electrodes have lower interfacial impedance than the traditional Ag/AgCl electrodes due to their excellent fluid deformability, which can adaptively fit the microtopography of the skin surface, and higher electrical conductivity. When the skin sweats, the interfacial impedance between the semi-liquid metal electrode and the skin decreases. This is because sweat forms a continuous ionic conductive film on the skin surface. It not only fills the tiny gaps between the electrode and the skin, but also penetrates the stratum corneum to increase its water content, and enhance internal ion migration to reduce resistance. Additionally, sweat acts as a lubricant and filling medium, promoting the semi-liquid metal to fully wet the skin, lowering contact resistance and thus the overall interfacial impedance.

Supporting information (Pages #12): **Fig. R31** and **R32** have been included as **Supplementary Fig. 45** and **Fig. 44** in the revised Supporting Information.

Comment 17. In supplementary figure 12, the wettability of the terminals is crucial for full functionality of the LED. Please refer to relevant literature(10.1016/j.snb.2015.07.062) and it is recommended to calculate current voltage across the LED after various modes of deformation to ascertain that no parasitic resistance is generated in the LM-LED interfaces.

Re: Thank you for your careful consideration of our manuscript. In the new manuscript, to reduce parasitic resistance, we pre-soaked the LEDs in a HCl solution to remove oxides from the pads and enable intermetallic wetting when the pads come into contact with the semi-liquid metal, thereby forming stable solder joints. Additionally, to reduce the parasitic resistance caused by deformation at the junction between the semi-liquid metal wire and the LED during stretching, we embedded rigid film into the VHB substrate to minimize the deformation at the junction, as shown in **Fig.R33a**. After these improvements, we measured the voltage and current across the LEDs during 200% stretching process, as presented in **Fig.R33b,c**. It can be observed from the figure that the voltage across the LED decreases under 200% stretching; this is because the resistance of the semi-liquid metal wire increases by 0.8 Ω. However, the increased wire resistance only causes a voltage drop of 4 mV and a current

drop of 4 μA across the LED, so the parasitic resistance at the junction between the LED pad and the semi-liquid metal wire is negligible.

Fig. R33 **a** LED on the VHB tape of composite rigid film under different stretching states. **b** The voltage across the LEDs during 200% stretching process. **c** The current across the LEDs during 200% stretching process.

The following changes have been introduced to address the comment:

Main text (line #370): Finally, to confirm its integration capability with traditional electronic devices, we used HCl solution to deoxidize the pins of traditional electronic components [46], such as LEDs, enabling them to be wetted and soldered with semi-liquid metal wires, forming stable electrical interconnections, as shown in **Supplementary Fig. 23**. Additionally, a rigid film (polyethylene) was embedded into the VHB tape to minimize deformation at the connection, as shown in **Supplementary Fig. 24a**. **Supplementary Fig. 24b, c** illustrate that during the 200% stretching process, the voltage and current across the LEDs gradually decreased, which is attributed to an increase of 0.8Ω in the resistance of the semi-liquid metal wire. However, the increased wire resistance only caused a voltage drop of 4 mV and a current reduction of $4 \mu\text{A}$ across the LEDs. Therefore, the variation in wire resistance induced by deformation has a negligible impact on the circuit function.

Main text (line #807):

[46] Li, G., Wu, X., Lee, D-W. Selectively Plated Stretchable Liquid Metal Wires for Transparent Electronics. *Sensors and Actuators B*. **221**, 1114-1119 (2015).

Supporting information (Pages #7): **Fig. R33** has been included as **Supplementary Fig. 24** in the revised Supporting Information.

Comment 18. In Figure 5e, what is the current value applied for heating?

Re: Thank you for your reminder. The current value applied for heating used in the experiment was 3A. In the new manuscript, we have supplemented the missing information.

Main text (line #522): As shown in Fig. 5E, the electrically heated wires can effectively heat the wing under a current of 3 A, increasing the surface temperature of the wing from 23.5°C to 60.7°C within 80 s.

REVIEWER #3

Comment: The authors present here an approach for using mechanical engraving in alcohol for high resolution patterning of liquid metal circuits. I thought overall the concept appears novel, testing appears thorough and the paper is largely well written with compelling figures; stretchable electronics and liquid metal circuits are also of substantial broader interest making the article well suited for your readership. There were however a number of minor errors and misstatements that should be corrected before acceptance; a full list of my concerns is below.

Re: We greatly appreciate the reviewer for such supportive comments. We have carefully considered each of your comments and suggestions. We have made extensive revisions to the manuscript, and we believe these changes have significantly improved the quality of our work. To help you quickly identify the modifications, we have highlighted the revised sections in red font throughout the document.

Comment 1. Fig. 3 C is more than a little confusing, appearing to conflate material property (the authors have a novel liquid metal loaded with copper particles, and get a higher conductivity) and fabrication approach. In practice, the two parameters are largely independent (the authors could for instance have injected their material into microchannels and gotten a different position on their plot) and it is misleading to imply that microchannel injection, or screen printing, or laser ablation, have specific conductivity numbers.

Re: Thank you for your careful consideration of our manuscript. We agree with your suggestions and have made the revisions. To avoid misleading readers into regarding the electrical conductivity of the bulk material as that of the wires, we have removed **Fig. 3c**. In fact, the resolution of the wires is independent of the materials own electrical conductivity, and the innovation of this study lies in the preparation method rather than the material itself. Therefore, the description about the innovation of the materials high electrical conductivity has also been removed from the new manuscript.

Comment 2. In Fig. 2 (a), water and common alcohols are well known materials and presumably contact angles on many common substrates are available in the literature, particularly substrates like PDMS that are very widely used in microfluidics. How did their results compare to the literature? Is this test truly necessary to include, given the common nature of the two liquids involved?

Re: Thank you for your careful consideration of our manuscript. We agree that contact angles of water and common alcohols on substrates such as PDMS have been reported in the literature. We have also conducted a search for relevant supporting literature: the water contact angle on PDMS measured in this study is $\sim 103^\circ$, and the anhydrous ethanol contact angle is $\sim 30^\circ$, which are highly consistent with previously reported values (e.g., [L. Shuai et al., Enhanced condensation heat transfer by water/ethanol binary liquids on polydimethylsiloxane brushes. *Droplet*, 2022]: water: $100^\circ \sim 105^\circ$; ethanol: $23^\circ \sim 31^\circ$). Given that the core objective of this study is not to investigate the contact angle characteristics of these two liquids on PDMS, the relevant data only serve as auxiliary information to describe the basic state of the substrate surface. By intuitively comparing the wettability differences between water and ethanol on a variety of common substrates, readers can clearly understand that ethanol enables the engraving method described in this paper more effectively than water.

Comment 3. In a scientific context, alcohol is a material class, not a specific liquid; in looking at the materials section the authors identify the material as athanol, which I am not familiar with and does not appear to be a common alcohol. Do the authors mean ethanol? If so, the authors should consistently and correctly identify this throughout the manuscript, rather than using alcohol

Re: We really appreciate the helpful comments from the reviewer. The "alcohol" mentioned throughout this study refers specifically to anhydrous ethanol (99%, purchased from Xintai Yixinkang Medical Supplies Co., Ltd.). In the new manuscript, we corrected "alcohol" to "ethanol" to ensure

reproducibility and clarity and corrected "Athanol" into "Anhydrous ethanol" in method section.

Comment 4. In the two cyclic tests (Figures 3 G and J), it is very important to clarify the bend and stretch conditions during the cycling; how much is the sample being bent or stretched in each case? It is somewhat concerning that the value never looks stable in either case (and for the stretching in particular the change is more than ten percent, which is not insubstantial).

Re: We highly appreciate the reviewers' critical reminders. In the revised manuscript, we have supplemented the missing experimental conditions, including the initial resistance, stretching conditions, and bending radius of the semi-liquid metal wires. We re-examined the semi-liquid metal wire samples used for tensile testing and found that the local agglomeration of solid metal particles caused a significant increase in resistance of the samples under high tensile conditions. Under high tensile stress, the narrowing of the semi-liquid metal wire width leads to the extrusion of agglomerated solid metal particles from the wire, resulting in a significant reduction in the cross-sectional area of the wire and thus a sharp increase in resistance. Therefore, in the new manuscript, we remanufactured the semi-liquid metal wires for tensile testing (**Fig. R34**) and added wire samples with widths of 0.5 mm and 2 mm to evaluate the effect of wire width on the electrical properties under tensile conditions (**Fig. R35**). The new samples were inspected using a microscope to ensure that there was no agglomeration of solid metal particles. We obtained the resistance variation of the three types of wires under the maximum stretching state, as well as the changes in their resistance during 10,000 cycles of stretching and bending. The experimental results are similar to those of the original semi-liquid metal wires (width: 1 mm): both can maintain stable connection of the wires without breakage. However, the resistance variation of the semi-liquid metal wires after cyclic stretching is relatively significant. The above results are similar to previous reports, both attributed to the fact that the stretchable substrate gradually relaxes after multiple stretches and fails to return to its initial length, coupled with the continuous formation of new oxides on the semi-liquid metal during repeated stretching, thus leading to a gradual increase in resistance. Although the resistance of the wire gradually increases with the increase in the number of stretching and bending cycles, this increase in resistance is relatively slight compared to that of commonly used electronic components. For example, the initial resistance of the semi-liquid metal wire is 1.65 Ω ; after 10,000 stretching cycles, its resistance increases by 20%, which is only an increase of 0.33 Ω -far lower than the internal resistance of commonly used electronic components. To verify this, we measured the voltage and current across the LEDs during 200% stretching process, as presented in **Fig.R36**. It can be observed from the figure that the voltage across the LED decreases under 200% stretching; this is because the resistance of the semi-liquid metal wire increases by 0.8 Ω . However, the increased wire resistance only causes a voltage drop of 4 mV and a current drop of 4 μ A across the LED, so the parasitic resistance at the junction between the LED pad and the semi-liquid metal wire is negligible.

Fig. R34 **f** Resistance variation of the three types of wires under different stretching states. **g** Resistance variation during 10,000 cyclic stretching cycles.

Fig. R35 **a** Resistance variation of semi-liquid metal wire (width of 2 mm) during 10,000 stretching cycles. **b** Relationship between resistance variation and bending angle during semi-liquid metal wire (width of 2 mm) bending. **c** Resistance variation of semi-liquid metal wire (width of 2 mm) during 10,000 bending cycles. **d** Resistance variation of semi-liquid metal wire (width of 0.5 mm) during 10,000 stretching cycles. **e** Relationship between resistance variation and bending angle during semi-liquid metal wire (width of 0.5 mm) bending. **f** Resistance variation of semi-liquid metal wire (width of 0.5 mm) during 10,000 bending cycles.

Fig. R36 **B** The voltage across the LEDs during 200% stretching process. **C** The current across the LEDs during 200% stretching process.

The following changes have been introduced to address the comment:

Main text (line #354): In addition, thinner (width: 0.5 mm) and thicker (width: 2 mm) semi-liquid metal wires were tested for stretching (**Fig. 3f**) and bending (**Supplementary Fig. 21**). Results showed the resistance variation of 2 mm-wide wire with initial resistance of 0.78Ω reached 9.5 at maximum stretch, and it withstood 10,000 cycles of 200% stretching without fracture. Similarly, the resistance variation of 0.5 mm-wide wire with initial resistance of 4.25Ω reached 13.2 at maximum stretch, and also endured 10,000 cycles of 200% stretching without breaking. For bending performance, the 2 mm-wide wire exhibited a resistance variation of only 1.004 at 90° bending and 1.04 after 10,000 bending cycles. Similarly, the 0.5 mm-wide wire showed 1.012 (90° bending) and 1.06 (10,000 cycles), respectively.

Main text (line #326): The bulk semi-liquid metal with initial resistance of 1.65Ω shows significant resistance differences depending on the change of geometric shape, and its resistance variation at a stretching rate of 1000% can reach 12.2. The semi-liquid metal wires exhibit reliable and stable responses under repeated 200% strain (more than 10,000 cycles), and no electrical failures occur (**Fig. 3g**).

Main text (line #370): Finally, to confirm its integration capability with traditional electronic devices, we used HCl solution to deoxidize the pins of traditional electronic components [46], such as LEDs,

enabling them to be wetted and soldered with semi-liquid metal wires, forming stable electrical interconnections, as shown in **Supplementary Fig. 23**. Additionally, a rigid film (polyethylene) was embedded into the VHB tape to minimize deformation at the connection, as shown in **Supplementary Fig. 24a**. **Supplementary Fig. 24b, c** illustrate that during the 200% stretching process, the voltage and current across the LEDs gradually decreased, which is attributed to an increase of 0.8Ω in the resistance of the semi-liquid metal wire. However, the increased wire resistance only caused a voltage drop of 4 mV and a current reduction of $4 \mu\text{A}$ across the LEDs. Therefore, the variation in wire resistance induced by deformation has a negligible impact on the circuit function.

Main text (line #678): In the stretching test, the VHB substrate was stretched to a 1000% stretching rate at a speed of 1mm/s; in the bending test, the PI substrate was bent to 90° with a bending radius of 5 mm, and one bending cycle takes 5 s.

Main text (line #807):

[46] Li, G., Wu, X., Lee, D-W. Selectively Plated Stretchable Liquid Metal Wires for Transparent Electronics. *Sensors and Actuators B*. **221**, 1114-1119 (2015).

Main text (line #382): **Fig. R34** has been included as **Fig. 3f, g** in the revised manuscript.

Supporting information (Pages #6-7): **Fig. R35, R36** have been included as **Supplementary Fig. 21** and **Fig. 24b, c** in the revised Supporting Information.

Comment 5. Based on the cycling test, it appears that even for a modest strain a permanent resistance change is occurring. The authors argue that there is value in their printed traces as a strain sensor up to nearly 1000% strain; is the resistance stable at higher strains under repeated strains? A more aggressive cycling strain test is necessary to defend this claim, based on what is being presented here.

Re: Your thorough review and valuable insights have been instrumental in refining our study, and we would like to express our sincere thanks. Considering the substantial variation in the resistance of the wires after cyclic stretching, we have removed the relevant description stating that the semi-liquid metal wires have the potential to be used as strain sensors from the revised manuscript. In addition, we performed 10,000 cycles of stretching on the wire samples under a higher strain (500%), as shown in **Fig. R37**. The results show that the wires can maintain stable connectivity, but the change in resistance is relatively obvious. Therefore, we suggest that this semi-liquid metal wire is suitable for use as a stretchable wire in flexible circuits, rather than directly as a strain sensor-this is because it requires resistance correction after multiple stretches.

Fig. R37 Resistance variation of semi-liquid metal wire under 500% strain during 10,000 stretching cycles.

The following changes have been introduced to address the comment:

Main text (line #347): Furthermore, the wire sample underwent 10,000 cyclic stretches at a higher

strain (500%). As shown in **Supplementary Fig. 20**, the wire remained conductive after 10,000 stretches under high strain, while its resistance variation increased to 5.83. These results align with previous reports, attributed to gradual relaxation of the stretchable substrate (failing to recover initial length) after repeated stretches, combined with continuous oxide formation on the semi-liquid metal during cyclic stretching, which both leading to progressive resistance increase. Thus, this semi-liquid metal wire is suitable for stretchable conductors in flexible circuits but not for direct use as strain sensors, as resistance correction is required after multiple stretches.

Supporting information (Pages #6): Fig. R37 have been included as **Supplementary Fig. 20** in the revised Supporting Information.

Comment 6. Recovering and recycling liquid metal electronics has been demonstrated previously elsewhere (for example, in L. Teng et al., Liquid metal-based transient circuits for flexible and recyclable electronics, *Adv. Funct. Mater.*, 2019), and needs to be properly credited and compared with here.

Re: Thank you for your reminder. We have added the literature (**Ref.S5**) you suggested to Table S1.

Table R1 Comparative framework across six key dimensions: manufacturing cost, recyclability, conductivity loss, material loss, resolution, and printability on diverse substrates.

	Manufacture re cost	Recyclable	Conductivity (S/m)	Material loss	Resolution(μm)	Printability on various substrates
This work	Low	√	9×10^6	×	High (5)	√
Ref.12	Low	×	$2.9 \times 10^5 - 1.2 \times 10^6$	×	Low (500)	√
Ref.19	High	×	3.4×10^6	√	High (20)	×
Ref.20	High	×	3×10^6	√	High(20)	×
Ref.21	High	√	3.4×10^6	√	High(0.18)	×
Ref.22	High	√	3.4×10^6	×	High(5)	√
Ref.23	Low	×	1.5×10^6	×	High(50)	√
Ref.26	High	√	2.06×10^6	×	High (25)	×
Ref.27	Low	√	3.4×10^6	√	Low (100)	×
Ref.28	High	×	3.4×10^6	√	350	×
Ref.29	High	×	5.65×10^5	√	High(4.5)	×
Ref.30	High	×	3.4×10^6	√	Low (150)	×
Ref.33	High	×	3.4×10^6	×	Low (200)	×
Ref.34	High	√	3.4×10^6	×	Low(200)	×
Ref.35	High	×	$3.4 - 6.73 \times 10^6$	√	Low (1.3)	×
Ref.36	Low	×	4.15×10^4	√	Low (100)	×
Ref.S1	High	√	3.4×10^6	×	High (1.9)	√
Ref.S2	High	×	$\sim 10^6$	√	Low (138)	×
Ref.S3	High	√	7.7×10^5	√	High(37)	×
Ref.S4	High	×	3.4×10^6	√	Low	×
Ref.S5	Low	√	-	×	Low (250)	×

[S1] Park, Y-G., An, H.S., Kim, J-Y., Park, J-U. High-resolution, reconfigurable printing of liquid metals with three-dimensional structures. *Sci. Adv.* **5**, eaaw2844(2019).

[S2] Li, S., Zhao, H., Xu, H., Lu, H., Luo, P., Zhou, T. Ultra-flexible stretchable liquid metal circuits with antimicrobial properties through selective laser activation for health monitoring. *Chem. Eng. J.* **482**, 149173 (2024).

- [S3] Liu, S., Kim, S.Y., Henry, K.E., Shah, D.S., Kramer-Bottiglio, R. Printed and Laser-Activated Liquid Metal-Elastomer Conductors Enabled by Ethanol/PDMS/Liquid Metal Double Emulsions. *ACS Appl. Mater. Interfaces*. **13**, 28729-28736 (2021).
- [S4] Li, Y., Feng, S., Cao, S., Zhang, J., Kong, D. Printable Liquid Metal Microparticle Ink for Ultrastretchable Electronics. *ACS Appl. Mater. Interfaces*. **12**, 50852-50859 (2020).
- [S5] L. Teng, S. C. Ye, S. Handschuh-Wang, X. H. Zhou, T. S. Gan, X. C. Zhou. Liquid Metal-Based Transient Circuits for Flexible and Recyclable Electronics. *Adv. Funct. Mater.* **29**, 1808739(2019).

The following changes have been introduced to address the comment:

Supporting information (Pages #12): **Table R1** have been included as **Table S1** in the revised Supporting Information.

Comment 7. Similarly, copper-liquid gallium amalgams do not appear to be new to this work, and have been previously investigated by these researchers (for instance in J. Tang et al., *ACS Appl. Mater. Interfaces* 2017). It is important to clarify this in the text to help the reader understand the specific contribution here.

Re: We greatly appreciate the reviewer for such supportive comments. These literatures have greatly helped improve the quality of our article. In the new manuscript, we have already cited these papers.

The following changes have been introduced to address the comment:

Main text (line #797):

[42] Tang, J. et al. Gallium-Based Liquid Metal Amalgams: Transitional-State Metallic Mixtures (TransM²ixes) with Enhanced and Tunable Electrical, Thermal, and Mechanical Properties. *ACS Appl. Mater. Interfaces*. **9**, 35977-35987 (2017).

REVIEWERS' COMMENTS

Reviewer #1 (Remarks to the Author):

Authors well modified the manuscript. The reviewer think that the paper is good for publication in Nat. Comm.

Re: We would like to express our sincere gratitude to the reviewer for your positive feedback and recognition of our revised manuscript. We greatly appreciate your acknowledgment that the manuscript, after revisions, meets the standards and is suitable for publication in Nature Communications.

Reviewer #2 (Remarks to the Author):

The authors have well addressed the prior concerns. I recommend publication of the revised manuscript.

Re: We would like to express our sincere gratitude to the reviewer for your positive feedback and recognition of our revised manuscript. We greatly appreciate your acknowledgment that the manuscript, after revisions, meets the standards and is suitable for publication in Nature Communications.

Reviewer #3 (Remarks to the Author):

The authors have addressed my concerns, and I believe the paper is ready for publication.

Re: We would like to express our sincere gratitude to the reviewer for your positive feedback and recognition of our revised manuscript. We greatly appreciate your acknowledgment that the manuscript, after revisions, meets the standards and is suitable for publication in Nature Communications.